# Phosphorylation regulates the binding of autophagy receptors to FIP200 Claw domain for selective autophagy initiation

Zixuan Zhou[1], Jianping Liu 📷 [1], Tao Fu[1], Ping Wu[2], Chao Peng 📷 [2], Xinyu Gong[1], Yingli Wang[1], Mingfang Zhang[1], Ying Li[1], Yaru Wang[1], Xiaolong Xu[1], Miao Li[3] & Lifeng Pan 📷 [1,3 ✉]

The ULK complex initiates the autophagosome formation, and has recently been implicated in selective autophagy by interacting with autophagy receptors through its FIP200 subunit. However, the structural mechanism underlying the interactions of autophagy receptors with FIP200 and the relevant regulatory mechanism remain elusive. Here, we discover that the interactions of FIP200 Claw domain with autophagy receptors CCPG1 and Optineurin can be regulated by the phosphorylation in their respective FIP200-binding regions. We determine the crystal structures of FIP200 Claw in complex with the phosphorylated CCPG1 and Optineurin, and elucidate the detailed molecular mechanism governing the interactions of FIP200 Claw with CCPG1 and Optineurin as well as their potential regulations by kinase-mediated phosphorylation. In addition, we define the consensus FIP200 Claw-binding motif, and find other autophagy receptors that contain this motif within their conventional LC3-interacting regions. In all, our findings uncover a general and phosphoregulatable binding mode shared by many autophagy receptors to interact with FIP200 Claw for autophagosome biogenesis, and are valuable for further understanding the molecular mechanism of selective autophagy.

[1] State Key Laboratory of Bioorganic and Natural Products Chemistry, Center for Excellence in Molecular Synthesis, Shanghai Institute of Organic Chemistry, University of Chinese Academy of Sciences, Chinese Academy of Sciences, Shanghai, China. [2] National Facility for Protein Science in Shanghai, Zhangjiang Lab, Shanghai Advanced Research Institute, Chinese Academy of Science, Shanghai, China. [3] School of Chemistry and Materials Science, Hangzhou Institute for Advanced Study, University of Chinese Academy of Sciences, Hangzhou, China. ✉email: panlf@sioc.ac.cn

Autophagy is a well regulated and critical intracellular catabolic process involving lysosome-dependent degradation of undesired cytosolic components, such as dysfunctional organelles, ubiquitinated bulk protein aggregates, and invasive pathogens, for cellular homeostasis and/or adaptation to various stresses in eukaryotic cells[1–4]. As a major subtype of autophagy in mammals, macroautophagy (hereafter referred to as autophagy) relies on the unique double-membraned vesicles named autophagosomes to encapsulate cytosolic materials for delivery to lysosomes[5,6]. In addition to the conventional non-selective "bulk" autophagy, recent studies have uncovered a substantial number of selective autophagy processes[7–13], such as the selective autophagy of aggregated proteins (aggrephagy)[14–17], invading pathogens (xenophagy)[18–20], dysfunctional mitochondria (mitophagy)[21–24], endoplasmic reticulum (ER) subdomains (ER-phagy)[25–30], glycogen (glycophagy)[31], and ferritins (ferritinophagy)[32]. During these selective autophagy processes, a unique type of adaptor proteins termed autophagy receptors are found to play an essential role[7–9,11,13]. So far, dozens of autophagy receptors have been identified in mammals, such as SQSTM1/P62, NBR1, Optineurin, CALCOCO2/NDP52, TAX1BP1, CCPG1, FAM134B, Nix, FUNDC1, and STBD1, all of which contain a cargo-associating domain or motif that can specifically recognize designated autophagic cargoes, and a LC3-interacting region (LIR) that can recruit ATG8 family proteins known as LC3A, LC3B, LC3C, GABARAP, GABARAPL1, and GABARAPL2 in mammals[7,8,11,13,33–37]. Thereby, autophagy receptors can specifically recognize autophagic cargoes and subsequently, induce in situ autophagosome formations by recruiting relevant autophagy machinery for the encapsulations of targeting cargoes and the ultimate autophagic degradation. In view of the crucial roles played by autophagy receptors in selective autophagy, the functions of autophagy receptors have been well-tuned temporally and spatially by other regulatory proteins, and defects in autophagy receptors or relevant regulatory proteins caused by gene mutations are associated with severe human diseases including neurodegenerative diseases and infectious diseases[7,8,11,13,38]. For instance, the TBK1 kinase can directly phosphorylate Optineurin, a crucial ubiquitin-binding autophagy receptor involved in xenophagy, aggrephagy and the depolarization-dependent mitophagy in mammals, to promote the efficiencies of Optineurin-mediated selective autophagy processes[19,36,39–41], and importantly, genetic mutations of Optineurin and TBK1 have been both linked with neurodegenerative diseases, such as amyotrophic lateral sclerosis (ALS)[42–44].

As a core molecular machinery in mammalian autophagy, the ULK complex is composed of a serine–threonine ULK1/2 kinase, ATG13, ATG101, and FIP200 (also named as RB1CC1) (Supplementary Fig. 1), and functions as a master regulator in the initiation of autophagy under the nutrient deprivation condition by linking the upstream nutrient sensors with the downstream autophagy machines[45–48]. Strikingly, recent studies revealed that the ULK complex also directly participates in the initiation of autophagosome formation in selective autophagy independent of nutrient status, and is required for several different types of selective autophagy processes including the CCPG1-mediated ER-phagy[30], P62-mediated aggrephagy[49], and NDP52-mediated xenophagy[50,51]. Specifically, during those ULK complex-involved selective autophagy processes, autophagy receptor can directly recruit the ULK complex to the vicinity of the targeting cargo for autophagy initiation by interacting with the FIP200 subunit[30,49–51], which is a large scaffold protein and mainly contains an ATG13-binding N-terminal domain, an LIR motif followed by several coiled-coil regions and an extreme C-terminal Claw domain (Supplementary Fig. 1). Interestingly, the ER-resident autophagy receptor CCPG1 and the cytosolic autophagy receptor P62 are reported to directly interact with the Claw region

of FIP200 through two short FIP200-interacting regions (FIR1 and FIR2) of CCPG1 and a region of P62 including the LIR motif, respectively[30,49]. In addition, the FIR regions of ATG16L1 and the TBK1-binding adaptors, NAP1 and SINTBAD, are also demonstrated to be recognized by the Claw region of FIP200[45,50,52]. However, due to the lack of related complex structures, the detailed molecular mechanisms underpinning the specific interactions between FIP200 and currently known FIP200-binding partners as well as the binding modes of the FIP200 Claw domain with FIR motifs are still largely unknown. Notably, the extreme C-terminal region of yeast ATG11 has some sequence similarity with the Claw domain of FIP200[47], and is implicated in the interaction with the yeast autophagy receptor ATG19 in a phosphorylation-dependent manner[53,54]. Whether the C-terminal Claw regions of FIP200 and yeast ATG11 share a conserved and phosphoregulatable binding mode to interact with autophagy receptors remains an open question. In addition, whether and how other autophagy receptors such as Optineurin interact with FIP200 are currently unknown.

In this work, we systemically characterize the interaction between FIP200 Claw domain and CCPG1 FIR2 motif, and find it can be regulated by the phosphorylation of a conserved Ser residue in the CCPG1 FIR2 region. In addition, we uncover that the LIR motif of Optineurin can be specifically recognized by the FIP200 Claw domain in a TBK1-mediated phosphorylation-dependent manner. The determined crystal structures of FIP200 Claw/phosphorylated CCPG1 FIR2 and FIP200 Claw/phosphorylated Optineurin LIR complexes not only elucidate the detailed molecular mechanism governing the interactions of FIP200 Claw with phosphorylated CCPG1 FIR2 and Optineurin LIR, but also uncover a similar and phosphoregulatable binding mode shared by CCPG1 and Optineurin for interacting with FIP200 Claw. Finally, based on our structural data, we define the consensus FIR core motif, and find a substantial number of autophagy receptors likely use the same motif for interacting with FIP200 Claw and ATG8 family proteins. Taken together, our findings provide mechanistic insights into the interactions of FIP200 with autophagy receptors CCPG1 and Optineurin, and expand our understandings of the interaction modes between FIR motifs and FIP200 Claw in general.

## Results

**Phosphorylation of the S104 residue in the FIR2 motif of CCPG1 enhances the interaction between CCPG1 FIR2 and FIP200 Claw domain.** To elucidate how the FIP200 Claw domain specifically recognizes its binding partners, we firstly focused on the interaction of the FIP200 Claw domain with the second FIR motif (FIR2) of CCPG1, and sought to determine their complex structure. However, after numerous trials, we failed to solve the complex structure using the wild-type CCPG1 FIR2 either by X-ray crystallography or NMR spectroscopy, likely due to the dynamic nature of this interaction. Interestingly, previous studies well demonstrated that the phosphorylation of Ser390 and Ser391 residues in the yeast autophagy receptor ATG19 can promote the interaction of ATG19 with the C-terminal region of ATG11, thereby initiating the Cvt pathway[53,54]. Therefore, we conducted a careful sequence alignment analysis of the currently known FIP200-binding regions of CCPG1, ATG16L1, NAP1, SINTBAD, and P62 from human species together with the ATG11-binding region of yeast ATG19 (residue 385–399), and revealed a consensus core sequence $\psi\psi\Theta xx\Gamma$ (where $\psi$ represents an acidic Asp residue or potentially phosphorylated Ser residue, $\Theta$ represents a bulk hydrophobic Ile or Trp residue, $\Gamma$ being a hydrophobic Leu or Ile residue, and x represents any residues) shared by those FIP200-binding proteins (Fig. 1a), suggesting that they may share

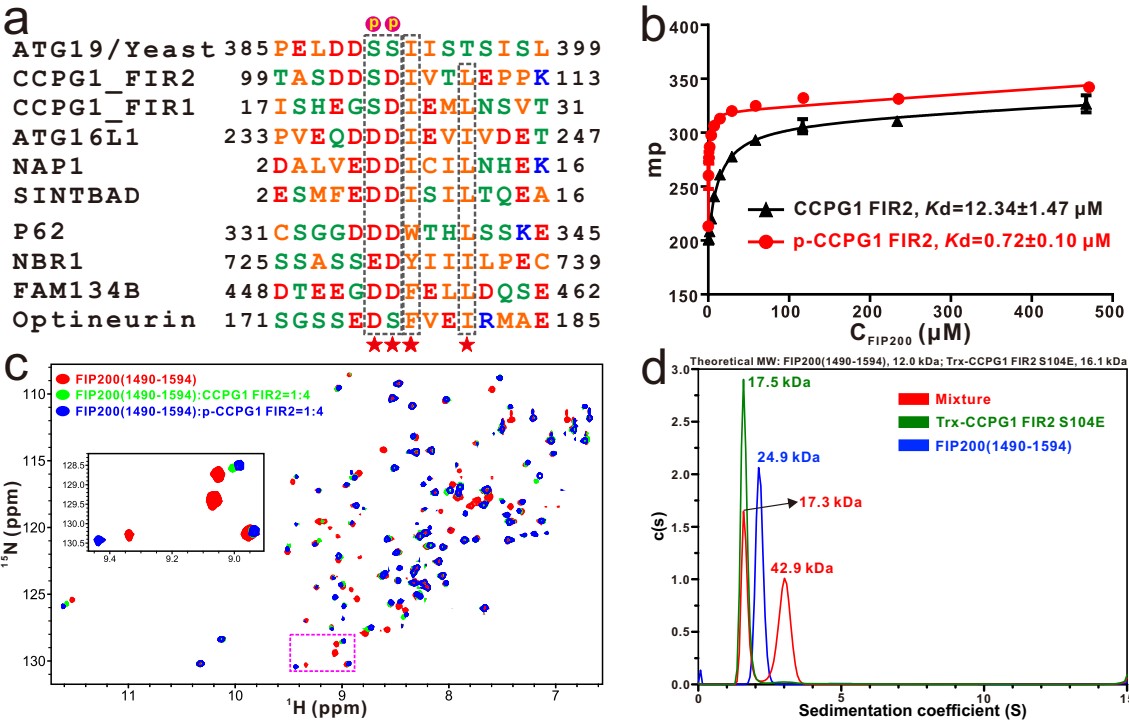

**Fig. 1 Sequence and biochemical analyses of the interaction of FIP200 Claw domain with CCPG1 FIR2. a** Sequence alignment analysis of the ATG11-binding motif of yeast ATG19 with the currently known FIP200 Claw-binding regions of CCPG1, ATG16L1, NAP1, and SINTBAD as well as the LIR regions of autophagy receptors, P62, NBR1, FAM134B, and Optineurin from human species. In this alignment, the highly conserved acidic residues (Asp, Glu or potentially phosphorylated Ser residue) and the following two conserved hydrophobic residues are boxed and highlighted with red stars, while the two phosphorylation sites of yeast ATG19 are further marked with "p". **b** Fluorescence polarization (FP) based assays measure the interactions of FIP200 Claw domain (residues 1490–1594) with CCPG1 FIR2 (residues 99–113) (black) and p-CCPG1 FIR2 (residues 99–113) (red). In this panel, p-CCPG1 FIR2 stands for the phospho-CCPG1 FIR2 peptide (SDDpSDIVTLEPPK). The $K_d$ values are the fitted dissociation constants with standard errors, when using the one-site binding model to fit the FP data. Experiments were performed in triplicate and error bars represented the standard deviation here ($n = 3$). Source data are provided as a Source Data file. **c** Superposition plots of the $^1$H-$^{15}$N HSQC spectra of the FIP200 Claw domain (red) and the protein titrated with CCPG1 FIR2 (green) or p-CCPG1 FIR2 (blue) at stoichiometric ratio of 1:4. For clarity, the insert shows the enlarged view of a selected region of the overlaid $^1$H-$^{15}$N HSQC spectra. **d** Overlay plot of the sedimentation velocity data of FIP200(1490–1594) (blue), the phosphomimetic CCPG1 FIR2 S104E mutant (green), and the FIP200(1490–1594) mixed with excess amounts of CCPG1 FIR2 S104E proteins (red). These results demonstrate that FIP200(1490–1594) forms a stable dimer, and can interact with the monomeric CCPG1 FIR2 S104E mutant to form a 2:2 stoichiometric complex in solution. Source data are provided as a Source Data file.

a similar binding mode to interact the Claw domain of FIP200. Importantly, further sequence conservation analysis together with the phosphorylation site predication by the NetPhos 3.1 server[55], suggested that the S104 residue located in CCPG1 FIR2 is a potential phosphorylation site (Supplementary Fig. 2b, c). Given that the extreme C-terminal regions of FIP200 and yeast ATG11 are highly conserved (Supplementary Fig. 3), we wondered whether phosphorylation of the FIR2 region of CCPG1 might regulate the interaction between CCPG1 and FIP200 Claw domain. To test this hypothesis, we quantitatively measured the interactions of FIP200 Claw (residue 1490–1594) with a 13-residue synthetic phospho-CCPG1 FIR2 peptide ("SDDpSDIV-TLEPPK", referred to as p-CCPG1 FIR2) and an un-phosphorylated CCPG1 FIR2 counterpart using fluorescence spectroscopy, and found the p-CCPG1 FIR2 binds to FIP200 Claw domain (residues 1490–1594) with a much stronger affinity compared with that of the wild-type CCPG1 FIR2 (Fig. 1b). Consistently, titrations of $^{15}$N-labeled FIP200(1490–1594) with un-labeled CCPG1 FIR2 or p-CCPG1 FIR2 showed that many peaks in the $^1$H-$^{15}$N HSQC spectrum of FIP200 Claw domain undergo distinct peak-broadenings or chemical shift changes in the presences of those two different FIR2 peptides (Fig. 1c and Supplementary Fig. 4), confirming that CCPG1 FIR2 can directly bind to FIP200 Claw domain and the phosphorylation of S104

residue in CCPG1 FIR2 can tune the binding of CCPG1 FIR2 with FIP200 Claw. In line with a previous report[49], further analytical ultracentrifugation-based assay revealed that FIP200 Claw forms a stable homodimer, which can simultaneously interact with two monomeric phosphomimetic S104E mutants of CCPG1 FIR2 to form a hetero-tetramer in solution (Fig. 1d).

**Overall structure of FIP200 Claw domain in complex with the phosphorylated CCPG1 FIR2.** Fortunately, using the purified FIP200(1490–1594) mixed with excess amount of p-CCPG1 FIR2 peptide, we successfully obtained high-quality crystals and solved the FIP200 Claw/p-CCPG1 FIR2 complex structure at 1.40 Å resolution (Supplementary Table 1). In an asymmetric unit, there is only one FIP200 Claw molecule that is bound with a p-CCPG1 FIR2 peptide in a 1:1 stoichiometry (Supplementary Fig. 5a). Further crystallographic symmetry analysis showed that the FIP200 Claw/p-CCPG1 FIR2 complex actually forms a symmetrical hetero-tetramer consisting of two p-CCPG1 FIR2 molecules and one FIP200 Claw dimer (Fig. 2a), in line with our analytical ultracentrifugation analysis (Fig. 1d). In the complex structure, the monomeric Claw domain of FIP200 adopts a unique architecture assembled by a 5-stranded anti-parallel β-sheet, a short α-helix, and a relatively isolated N-terminal β0-strand that

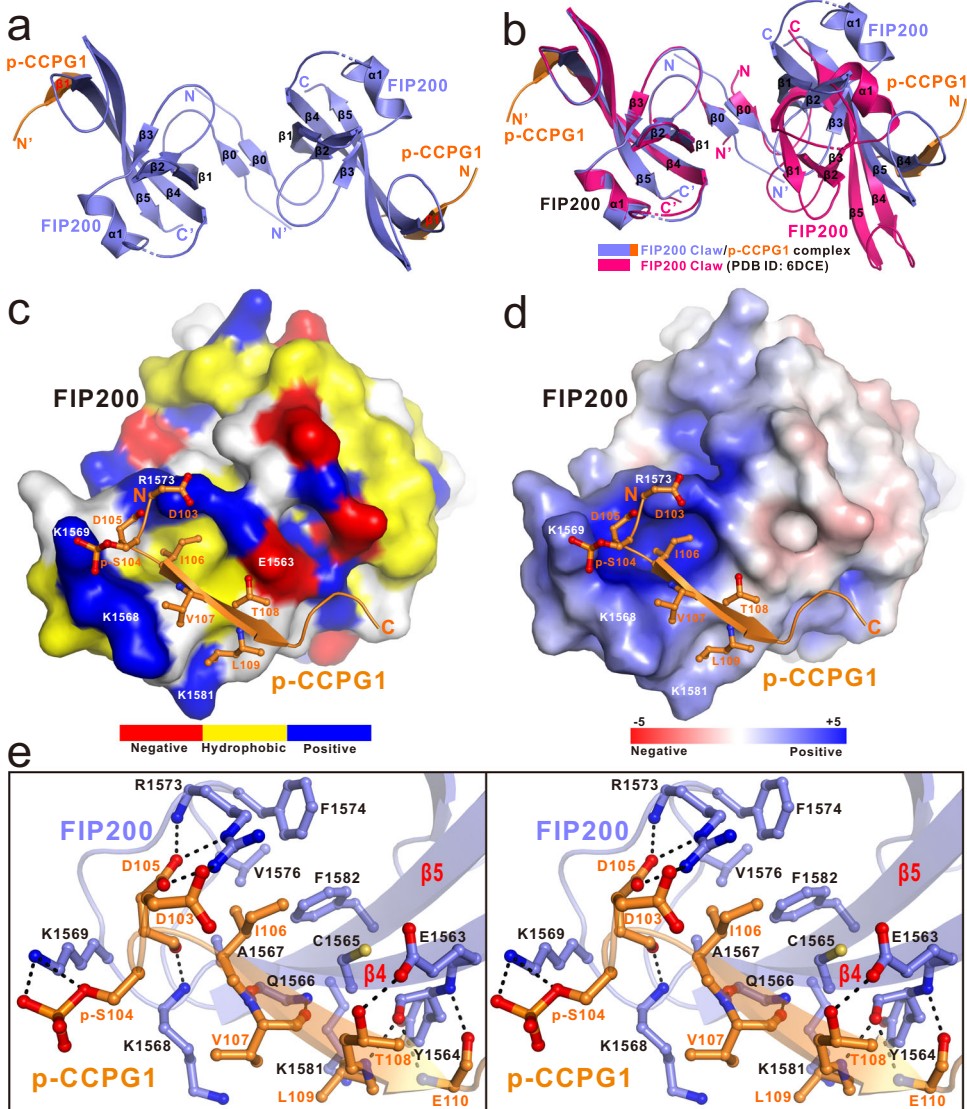

**Fig. 2 The structure of FIP200 Claw/p-CCPG1 FIR2 complex. a** Ribbon diagram showing the overall structure of the dimeric FIP200 Claw/p-CCPG1 FIR2 complex. In this drawing, p-CCPG1 stands for the phospho-CCPG1 FIR2 motif. Meanwhile, two FIP200 Claw domains are colored in slate, while the two bound p-CCPG1 FIR2 molecules are colored in orange, and meanwhile. **b** The comparison of the overall structures of the apo-form FIP200 Claw dimer (hot pink, PDB ID: 6DCE) and the FIP200 Claw/p-CCPG1 FIR2 complex (slate-orange). **c** The combined surface representation and the ribbon-stick model showing the hydrophobic binding interface between FIP200 Claw and p-CCPG1 FIR2. In this drawing, the p-CCPG1 FIR2 is displayed in the ribbon-stick model, and the FIP200 Claw domain is shown in surface representation colored by amino acid types. Specifically, the hydrophobic amino acid residues in the surface model of FIP200 Claw are drawn in yellow, the positively charged residues in blue, the negatively charged residues in red, and the uncharged polar residues in gray. **d** The combined surface charge potential representation (contoured at ±5 kT/eV; blue/red) and the ribbon-stick model showing the charge–charge interactions between FIP200 Claw and p-CCPG1 FIR2 in the complex structure. **e** Stereo view of the ribbon-stick model showing the detailed interactions between the Claw domain of FIP200 and p-CCPG1 FIR2. The related hydrogen bonds and salt bridges involved in the binding are shown as dotted lines.

directly packs with the counterpart β0-strand of another Claw monomer in an anti-parallel manner for the homo-dimerization of Claw (Fig. 2a and Supplementary Fig. 5a). The clearly defined p-CCPG1 FIR2 in the complex structure contains 11 highly conserved residues (DpSDIVTLEPPK) (Supplementary Figs. 2b and 5b), and mainly forms a short β-strand that directly augments the C-terminal portion of the β4-strand of FIP200 Claw in an anti-parallel manner and at a location far away from the Claw dimerization interface (Fig. 2a). Intriguingly, further structural comparison analysis of the dimeric FIP200 Claw/p-CCPG1 FIR2 complex with the apo-form FIP200 Claw dimer (PDB ID: 6DCE) revealed that although the monomeric FIP200 Claw domains in

the two structures are highly similar (Supplementary Fig. 6a), the overall conformations as well as the assemblies of the FIP200 Claw dimers are quite different (Fig. 2b and Supplementary Fig. 7a–d). In particular, the intermolecular hydrophobic contacts for the homo-dimerization of the FIP200 Claw domain in these two structures are totally different (Supplementary Fig. 8a, b). Moreover, the dimerization of the FIP200 Claw domain in the FIP200 Claw/p-CCPG1 FIR2 complex is mediated by more additional polar interactions, such as the two Arg–Asp pairs (Arg1491–Asp1500) and two Arg-Glu pairs (Arg1499-Glu1494) of salt bridges (Supplementary Fig. 8a, b), and buries a much larger surface area compared with that of the apo-form FIP200

Claw (~786 vs ~730 Å$^2$) (Supplementary Fig. 7a–d). Overall, the binding of p-CCPG1 FIR2 to the FIP200 Claw dimer is likely to open up a hydrophobic pocket located between two monomeric FIP200 Claw molecules (Supplementary Fig. 7b, d). Interestingly, further structural comparisons of our FIP200 Claw/p-CCPG1 FIR2 complex structure with the previously determined structure of FIP200 C-terminal fragment including the Claw domain and the preceding coiled-coil domain (PDB ID: 6GMA), revealing that the rearrangement of the FIP200 Claw dimer induced by p-CCPG1 FIR2 binding may cause large conformational changes of the coiled-coil region preceding the Claw of FIP200 (Supplementary Fig. 9).

**The molecular interface of the phosphorylated CCPG1 FIR2 and FIP200 Claw interaction.** In the FIP200 Claw/p-CCPG1 FIR2 complex structure, the entire p-CCPG1 FIR2 packs extensively with a solvent-exposed concave groove formed by the β4 and β5 strands as well as the β4/β5 connecting loop that folds back to the β-sheet of FIP200 Claw, burying a total surface area of ~579 Å$^2$ (Fig. 2a, c, d). Further detailed structural analyses of the binding interface of the FIP200 Claw/p-CCPG1 FIR2 complex revealed that the interaction between FIP200 Claw and p-CCPG1 FIR2 is mainly mediated by extensive polar (charge–charge and hydrogen bonding) and hydrophobic interactions (Fig. 2c–e). In particular, the negatively charged D103, D105 residues, and p-S104 of p-CCPG1 FIR2 form specific charge–charge and hydrogen bonding interactions with the positively charged R1573 and K1569 residues located in the β4/β5 connecting loop of FIP200 Claw, respectively (Fig. 2d, e). Moreover, the backbone groups of D105, V107, T108, and E110 residues located at the short β-strand region of p-CCPG1 FIR2 interact with the backbone groups of K1568, Q1566, Y1564 residues of FIP200 to form six strong backbone hydrogen bonds, and the side-chain hydroxyl group of T108 forms a specific hydrogen bond with the side chain of E1563 located at the β4 of FIP200 Claw (Fig. 2e). In addition, the hydrophobic side chain of CCPG1 I106 occupies a large hydrophobic pocket (LHP) formed by the side chains of C1565, A1567, F1574, V1576, and F1582 from FIP200, and concurrently, the hydrophobic side chain of CCPG1 L109 residue packs against a small hydrophobic groove (SHG) formed by the side chain of Y1564 and the aliphatic side chain of K1581 from FIP200 (Fig. 2c, e). Meanwhile, the hydrophobic side chain of CCPG1 V107 residue has a hydrophobic contact with the aliphatic side chain of FIP200 K1568 (Fig. 2e). In line with their important structural roles, all of these key binding interface residues of CCPG1 and FIP200 are highly conserved across different vertebrates (Supplementary Fig. 2a, b). It is worth noting that the critical positively charged K1569 and R1573 in FIP200 Claw are not conserved in the yeast ATG11 (Supplementary Fig. 3), therefore the C-terminal Claw-like region of yeast ATG11 may adopt a different mode to interact with its binding partners. Using analytical gel filtration chromatography and fluorescence spectroscopy-based analyses, we further verified the specific interactions between FIP200 Claw and CCPG1 FIR2 observed in the complex structure. In accordance with our aforementioned structural data, the results showed that point mutations of key interface residues either from CCPG1 FIR2 or FIP200 Claw, such as the D105A, I106S, T108A, L109A mutations of CCPG1 FIR2, or the Y1564S, K1569A, R1573E, F1574Q mutations of FIP200 Claw, all significantly decrease or essentially disrupt the specific interaction between CCPG1 FIR2 and FIP200 Claw (Supplementary Figs. 10a, c–f, 11, and 12). In contrast, the substitution of S104 with a phosphomimetic Glu residue in CCPG1 FIR2 enhances the interaction of CCPG1 FIR2 with FIP200 Claw (Supplementary Fig. 10a, b).

**TBK1-mediated phosphorylation of Optineurin S177 enhances the interaction between Optineurin LIR region and FIP200 Claw.** Interestingly, we also determined an apo-form crystal structure of the FIP200 Claw dimer (Supplementary Table 1), in which the highly positively charged LHP region of one Claw monomer is occupied by an aromatic Phe residue together with a preceding negatively charged Glu residue from the remaining sequences of the N-terminal cleaved 3C protease site of a neighboring FIP200 Claw dimer, likely due to crystal packing (Supplementary Fig. 13). Inspired by these structural observations, we inferred that an aromatic residue (Phe, Tyr, or Trp) corresponding to the I106 residue of CCPG1 FIR2 may also occupy the LHP of FIP200. Given that many canonical LIR motifs share a signature ψΘxxΓ (where ψ represents an acidic residue, Θ represents a bulk aromatic Phe, Tyr, or Trp residue, Γ being a hydrophobic Leu, Ile, or Val residue, and x represents any residues) sequence, which is similar to the core motif of FIR uncovered by the determined p-CCPG1 FIR2/FIP200 Claw complex structure (Figs. 1a and 2e). Notably, the LIR region of P62 was recently demonstrated to participate in the interaction with FIP200 Claw, and this interaction can be further enhanced by the phosphorylation of four residues located in the LIR region of P62[49]. Therefore, we inferred that other autophagy receptors might also directly interact with FIP200 Claw through their LIR regions and regulated by relevant kinase-mediated phosphorylation. To test our hypothesis, we focused on the potential interaction between FIP200 and Optineurin. Intriguingly, previous our studies together with other people's reports revealed that the TBK1 kinase can directly interact with Optineurin and regulate the functions of Optineurin in selective autophagy[19,36,39–41]. Strikingly, our biochemical analyses showed that TBK1 can directly phosphorylate the Optineurin(33–209) fragment that includes the N-terminal TBK1-binding coiled-coil and the LIR region of Optineurin (Fig. 3a and Supplementary Fig. 14), and importantly, the phosphorylated Optineurin(33–209) fragment displays a much stronger binding ability to FIP200 compared with the wild-type Optineurin(33–209) (Fig. 3b, c). Consistent with previous studies[19,39], our mass spectrometry-based analysis confirmed that the S177 residue preceding the canonical LIR core sequence (FEVI) of Optineurin is the major phosphorylated site mediated by TBK1 in the Optineurin(33–209) fragment (Fig. 3d). Using analytical gel filtration chromatography-based analyses, we further narrowed down the FIP200-binding region of Optineurin (33–209) to its LIR region, as the phosphomimetic S177E mutant of Optineurin(169–185) that solely includes the LIR motif of Optineurin, can specifically interact with FIP200(1450–1594) (Supplementary Figs. 14 and 15a), while the Optineurin(33–168) and Optineurin(186–209) fragments have no obvious interactions with FIP200 (Supplementary Fig. 15b, c). Consistently, using quantitative fluorescence spectroscopy-based assays, we demonstrated that a 13-residue synthetic S177-phosphorylated Optineurin LIR peptide ('SSEDpSFVEIRMAE', referred to as p-Optineurin LIR) can directly bind to FIP200 Claw with a $K_d$ value of ~12 μM, which is ~27-fold stronger than that of the unphosphorylated Optineurin LIR peptide ($K_d$ ~307 μM) (Fig. 3e). Taken together, all the above biochemical results clearly demonstrated that TBK1-mediated phosphorylation of S177 in the LIR region of Optineurin can dramatically promote the interaction of Optineurin LIR region with FIP200 Claw. As expected, the further substitution of the crucial F178 residue in the phosphomimetic S177E mutant of Optineurin LIR with an aromatic Tyr or Trp residue retained its ability to interact with FIP200 Claw (Supplementary Fig. 15d, e), indicating that other similar types of LIR motifs with an aromatic Tyr or Trp residue at the corresponding F178 of Optineurin LIR found in other autophagy receptors, such as the LIR motifs of P62, NBR1 and

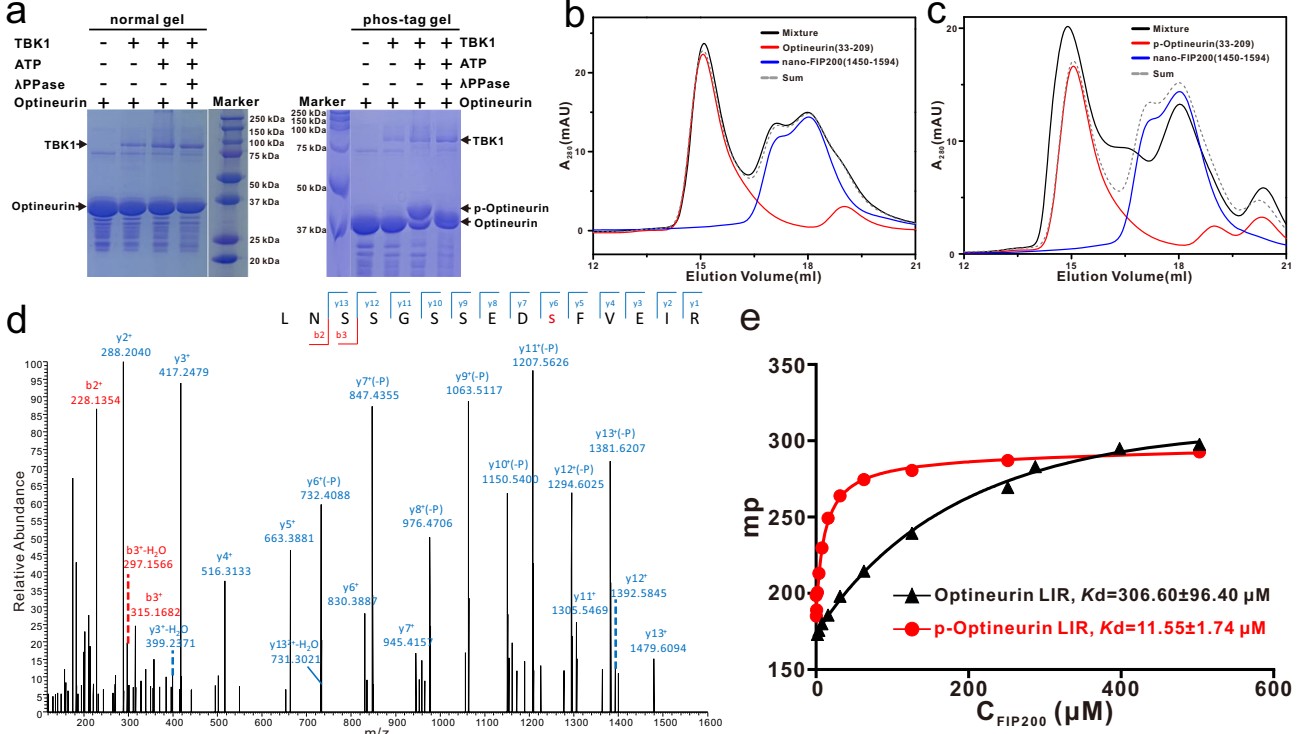

**Fig. 3 TBK1-mediated phosphorylation of the LIR region of Optineurin can promote the interaction between Optineurin and FIP200 Claw. a** In vitro phosphorylation assays showing that Optineurin(33–209) could be specifically phosphorylated by the purified TBK1 kinase. The gel in the left panel is normal SDS-PAGE gel, while the right one is a phos-tag gel. In this panel, p-Optineurin stands for phosphorylated Optineurin(33–209). This experiment was repeated twice independently with similar results. Source data are provided as a Source Data file. **b** Analytic gel filtration chromatography-based analysis of the interaction of FIP200 Claw domain with Optineurin(33–209). In this panel, "Sum" stands for the theoretical sum of Optineurin(33–209) and nano-FIP200(1450–1594) profiles, while "Mixture" stands for the Optineurin(33–209) and nano-FIP200(1450–1594) mixture sample. Source data are provided as a Source Data file. **c** Analytic gel filtration chromatography-based analysis of the interaction of FIP200 Claw domain with the phosphorylated Optineurin(33–209) mediated by TBK1, which is named as p-Optineurin(33–209). Source data are provided as a Source Data file. **d** The figure shows a higher-energy collisional dissociation (HCD) MS/MS spectrum recorded on the [M+2H]$^{2+}$ ion at $m/z$ 853.87 of the human Optineurin peptide LNSSGGSSEDsFVEIR harboring one phosphorylated site (denoted by lowercase s). Predicted b- and y-type ions (not including all) are listed above and below the peptide sequence, respectively. Ions observed are labeled in the spectrum and indicate that the S177 residue of Optineurin(33–209) protein is modified with the phosphate. **e** Fluorescence polarization assays measure the binding affinities of FIP200 Claw domain (residues 1490–1594) with Optineurin LIR (residues 169–185) (black) and the phosphorylated Optineurin LIR (p-Optineurin LIR) (red). $K$d values are the fitted dissociation constants with standard errors when using the one-site binding model to fit the FP data. Source data are provided as a Source Data file.

FAM134B (Fig. 1a), should also be recognized by FIP200 Claw. Indeed, the LIR motif of P62 was recently demonstrated to participate in the interaction with the FIP200 Claw domain[49].

**The structure of FIP200 Claw in complex with the phosphorylated Optineurin LIR.** To further uncover the detailed interaction mode between FIP200 Claw and p-Optineurin LIR, we also determined the high-resolution crystal structure of the FIP200 Claw/p-Optineurin LIR complex (Supplementary Table 2). As expected, the structure of the FIP200 Claw/p-Optineurin LIR complex is composed of a symmetric FIP200 Claw dimer and two p-Optineurin LIR molecules, each of which mainly forms a short β-strand and directly packs with the β4-strand of FIP200 Claw domain in an anti-parallel fashion (Fig. 4a). The overall binding mode of p-Optineurin LIR to the FIP200 Claw domain is very similar to that of p-CCPG1 FIR (Figs. 2a and 4a). In addition, the overall structure of the monomeric FIP200 Claw in the FIP200 Claw/p-Optineurin LIR complex structure is also highly similar to that of the monomeric FIP200 Claw domain in the apo-form structure (PDB ID: 6DCE) and in the FIP200 Claw/p-CCPG1 FIR2 complex structure (Supplementary Fig. 6b, c). Interestingly, in contrast to the dramatic conformation arrangement of the FIP200 Claw dimer induced by the p-CCPG1

FIR-binding (Fig. 2b), the binding of p-Optineurin LIR to FIP200 Claw only causes some relatively mild conformational changes of the FIP200 Claw dimer (Fig. 4b), and accordingly, the dimerization interface of FIP200 Claw dimer in the FIP200 Claw/p-Optineurin LIR complex more resembles that of the apo-form FIP200 Claw dimer (Supplementary Figs. 7 and 8).

Further structural characterizations showed that the specific interaction between FIP200 Claw and p-Optineurin LIR is mainly mediated by hydrophobic contacts and polar interactions (Fig. 4c–e). Particularly, the aromatic side chain of Optineurin F178 occupies the LHP of FIP200 Claw, and forms a unique cation-π interaction with the positively charged FIP200 R1584 residue (Fig. 4c–e). Meanwhile, the hydrophobic side chains of V179 and I181 pack against the aliphatic side chain of K1568 and the SHG of FIP200, respectively (Fig. 4c, e). Moreover, the backbone groups of phosphorylated S177 (p-S177), V179, and I181 form four strong backbone hydrogen bonds with the K1568, Q1566, and Y1564 residues of FIP200 (Fig. 4e). Similar to that of p-CCPG1 FIR2 in binding to FIP200 Claw (Fig. 2c–e), the highly positively charged K1569 and R1573 residues decorated around the LHP of FIP200 Claw are neutralized by three negatively charged E175, D176, and p-S177 residues from p-Optineurin LIR (Fig. 4c–e). However, in contrast to that of p-CCPG1 FIR2, the

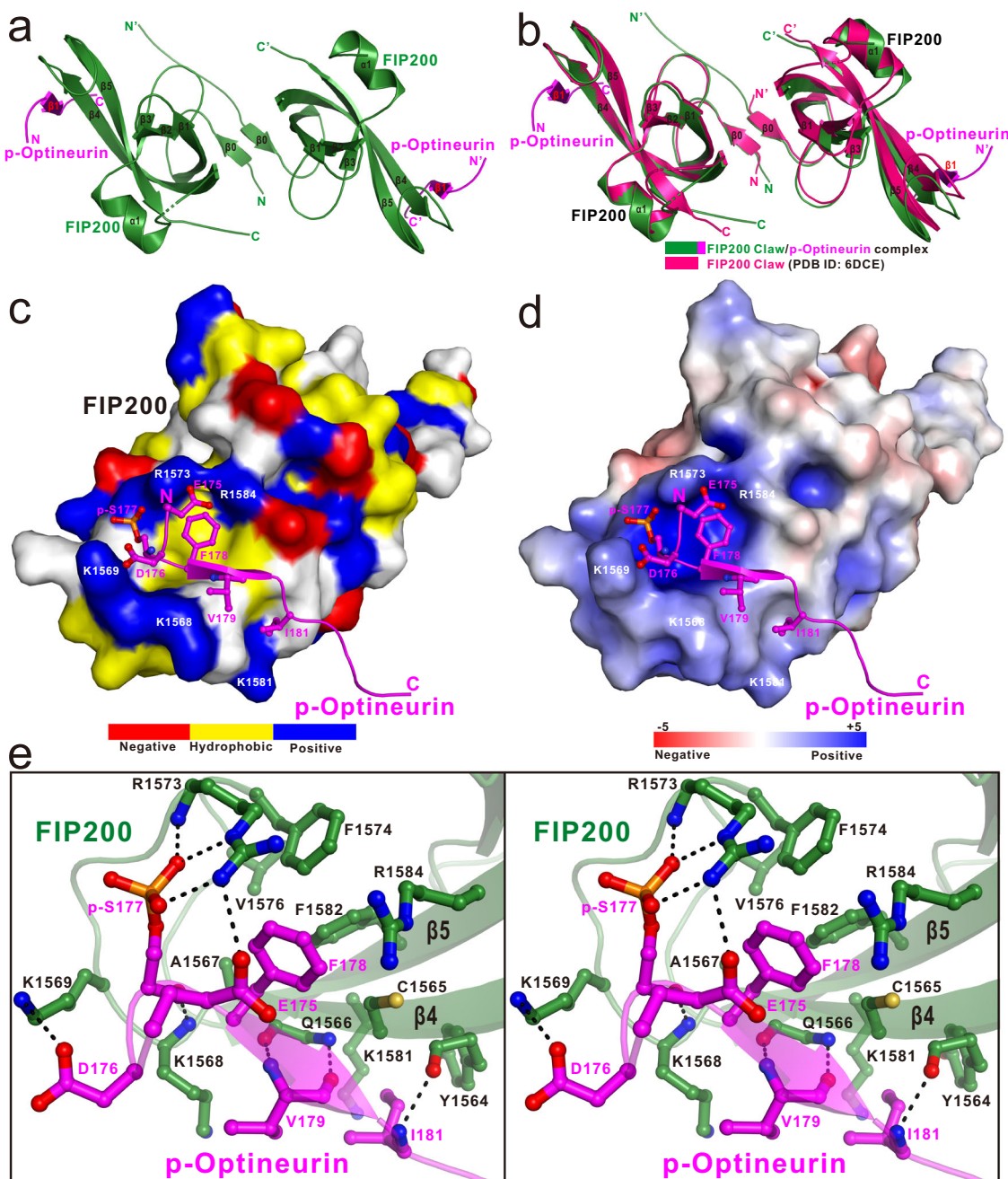

**Fig. 4 The structural analyses of FIP200 Claw/p-Optineurin LIR complex. a** Ribbon diagram showing the overall structure of the dimeric FIP200 Claw/p-Optineurin LIR complex. In this drawing, p-Optineurin stands for the phospho-Optineurin LIR motif. Meanwhile, the two FIP200 Claw molecules are colored in green, while the two bound p-Optineurin LIR are colored in magenta. **b** The comparison of the overall structures of the FIP200 Claw/p-Optineurin LIR complex (green-magenta) with the apo-form FIP200 Claw dimer (hot pink, PDB ID: 6DCE). **c** The combined surface representation and the ribbon-stick model showing the hydrophobic binding surface between FIP200 Claw and p-Optineurin LIR. In this drawing, the p-Optineurin LIR is displayed in the ribbon-stick model, and the FIP200 Claw domain is showed in surface representation colored by amino acid types. Specifically, the hydrophobic amino acid residues in the surface model of FIP200 Claw are drawn in yellow, the positively charged residues in blue, the negatively charged residues in red, and the uncharged polar residues in gray. **d** The combined surface charge representation and the ribbon-stick model showing the charge–charge interactions between FIP200 Claw and p-Optineurin LIR in the complex structure. **e** Stereo view of the ribbon-stick model showing the detailed interactions between the Claw domain of FIP200 and p-Optineurin LIR. The hydrogen bonds and salt bridges involved in the interaction are shown as dotted lines.

phosphate group of the acidic p-S177 preceding the crucial hydrophobic residue that occupies the LHP of FIP200 Claw is coupled with the FIP200 R1573 residue by forming two specific salt bridges and one hydrogen bond, while the side-chain carboxyl group of Optineurin D176 forms a salt bridge with the side chain of K1569 from FIP200 (Fig. 4c–e). In addition, a charge–charge interaction between Optineurin E175 and FIP200

R1573 further contributes to the p-CCPG1 FIR2/FIP200 Claw complex formation (Fig. 4d, e). Importantly, in line with our structural data, further analytic gel filtration chromatography-based analyses revealed that point mutations of key interface residues including the Y1564S, K1569A, R1573E, F1574Q mutations of FIP200 and the D176R, F178Q mutations of the phosphomimetic Optineurin(169–185) S177E mutant all

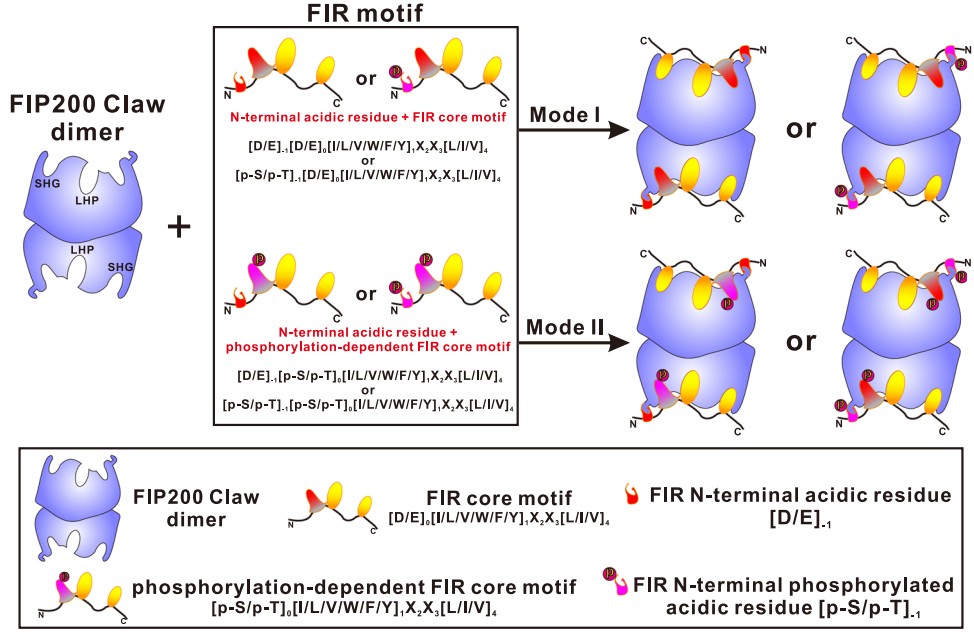

**Fig. 5 A schematic cartoon diagram summarizing the two different binding modes of FIR motifs in interacting with the FIP200 Claw domain.** In this diagram, "LHP" stands for the large hydrophobic pocket of FIP200 Claw formed by the side chains of C1565, A1567, F1574, V1576, and F1582 from FIP200, and "SHG" stands for the small hydrophobic groove of FIP200 Claw formed by the side chains of FIP200 Y1564 and K1581.

significantly attenuate or completely abolish the specific interaction between FIP200 and the phosphomimetic Optineurin (169–185) S177E mutant (Supplementary Fig. 16).

**Definition of the consensus FIR core motif.** Notably, the phosphomimetic S104E mutant of CCPG1 FIR2 and S177E mutant of Optineurin LIR region exhibited good abilities to interact with FIP200 Claw (Supplementary Figs. 10b and 15a), suggesting that FIP200 Claw domain can also recognize target proteins in a phosphorylation-independent manner if the residue corresponding to the S104 of CCPG1 or the S177 of Optineurin in such target proteins is a negatively charged Glu or Asp residue. Indeed, sequence alignment analysis showed that the FIR motifs of ATG16L1, NAP1, and SINTBAD all contain a negatively charged Asp residue in the corresponding CCPG1 S104 position, and the potential FIP200 Claw-binding motifs of P62, NBR1, and FAM134B all include a negatively charged Asp residue in the corresponding Optineurin S177 site (Fig. 1a). Furthermore, structural modeling analyses revealed that the LIR regions of P62, NBR1, and FAM134B may adopt a highly similar binding mode to interact with the FIP200 Claw domain as that of p-CCPG1 FIR2 or p-Optineurin LIR (Supplementary Fig. 17). Given that similar key FIP200 Claw-binding interface residues of CCPG1 FIR2 or Optineurin LIR can be also found in the FIR motifs of ATG16L1, NAP1, and SINTBAD as well as the LIR motif region of P62 (Fig. 1a), therefore these currently known FIP200-binding proteins including CCPG1, ATG16L1, NAP1, SINTBAD, P62, and Optineurin likely share a general binding mode to interact with the FIP200 Claw domain. Finally, based on our studies, we proposed a consensus FIR core motif for binding to FIP200 Claw: $\psi\Theta$xx$\Gamma$, where $\psi$ represents an acidic Asp, Glu or phosphorylated Ser/Thr residue, $\Theta$ represents a bulk hydrophobic Ile, Leu, Val or aromatic Phe, Tyr, Trp residue, $\Gamma$ being a hydrophobic Leu, Ile or Val residue, and x represents any residues. Apparently, this FIR core motif is essential for the interaction with FIP200 Claw, and can further work together with additional N-terminally adjacent acidic residues and/or serine/threonine phosphorylation sites for

the effective binding to FIP200 Claw. In addition, we categorized the currently known and the potential FIR-mediated target/FIP200 Claw interactions into two modes. Mode I, which was observed in the p-CCPG1 FIR2/FIP200 Claw complex (Fig. 2c–e), involved the synergic bindings of a phosphorylation-independent FIR core motif and an preceding acidic motif (either negatively charged residues or phosphorylated serine/threonine sites) to the FIR-docking site of FIP200 Claw (Fig. 5). Mode II, which was observed in the p-Optineurin LIR/FIP200 Claw interaction (Fig. 4c–e), required a phosphorylation-dependent FIR core motif and relied on its synergic binding with a preceding acidic motif (either negatively charged residues or phosphorylated serine/threonine sites) to the FIP200 Claw (Fig. 5). Notably, the phosphorylation of the FIR core motif is essential for the target/FIP200 Claw interaction in Mode II, while the phosphorylation of serine/threonine residue immediately preceding the FIR core motif in Mode I or II can further enhance the target/FIP200 Claw interaction.

**Discussion**

Previous studies have well established a critical role of the FIP200 subunit in the ULK complex for associating with other autophagic factors, especially some autophagy receptors such as CCPG1, NDP52, and P62[30,49–51]. In particular, the detailed characterization of FIP200 and P62 interaction by Turco et al.[49], revealed that a disordered region (residues 326–380) including the LIR motif of P62 can directly bind to the Claw domain of FIP200, and the phosphorylation of four residues (S349, S365, S366, and S370) C-terminally to the core LIR motif of P62 can further enhance the interaction between P62 and FIP200 Claw[49]. They determined the apo-form dimeric structures of FIP200 Claw and a FIP200 C-terminal fragment containing the Claw as well as the preceding coiled-coil domain[49], but, unfortunately, due to the lack of P62/FIP200 Claw complex structure, how FIP200 Claw interacts with P62 are still largely known. In this study, we determined two high-resolution crystal structures of the FIP200 Claw domain in complex with the phosphorylated CCPG1 FIR2

motif and Optineurin LIR, and presented the first atomic pictures showing how FIP200 associates with relevant autophagy receptors through its Claw domain. Importantly, our studies revealed that like that of FIP200 Claw and P62 interaction[49], the specific interactions of FIP200 Claw with CCPG1 and Optineurin can be regulated and enhanced by kinase-mediated phosphorylation in their respective FIP200-binging motifs (Figs. 1b and 3), although the relevant kinase, as well as the upstream signaling cascade for the phosphorylation of CCPG1 FIR2, remains to be elucidated in the future. Notably, the canonic LIR motif of P62 was well demonstrated to directly engage in the interaction with FIP200 Claw[49], and belongs to the Mode I type of FIR defined in this study with a phosphorylation-independent FIR core motif and two preceding acidic Asp residues (Figs. 1a and 5). However, in contrast to that of CCPG1 and Optineurin, the phosphor-egulatable sites of P62 are located within a C-terminal region following the LIR motif, and the phosphomimetic quadruple mutations (S349D, S365D, S366D, and S370D) of P62 can only enhance the binding affinity of P62 for FIP200 Claw roughly by twofold[49]. Therefore, how these phosphoregulatable sites of P62 interact with FIP200 Claw remains to be elucidated. Interestingly, comparisons of our determined complex structures with the previously solved apo-form structure of FIP200 Claw (PDB ID: 6DCE) revealed that the overall structures of the monomeric FIP200 Claw domains in these structures are essentially the same (Supplementary Fig. 6), however, the bindings of p-CCPG1 FIR2 and p-Optineurin LIR to FIP200 Claw dimer can induce different conformational changes (Figs. 2b and 4b). Especially, the relatively stronger binding of p-CCPG1 FIR2 to FIP200 Claw can cause a large conformational arrangement of the dimeric FIP200 Claw domains (Fig. 2b and Supplementary Figs. 7, 9a). Given that the activity of the ULK complex is essential for autophagosome nucleation, whether such conformational changes of the dimeric FIP200 Claw might potentially regulate the ability of FIP200 or other related regulatory protein to activate ULK1/2 for initiating autophagosome formation still awaits further investigation.

A previous study from Richard J. Youle's group well demonstrated that the function of Optineurin in Parkin-mediated mitophagy involves the recruitment of the ULK complex[24]. Consistently, in this study, we demonstrated that Optineurin can directly interact with the FIP200 subunit of ULK complex through its LIR region in a TBK1-mediated phosphorylation-dependent manner (Fig. 3). Particularly, the phosphorylation of S177 residue within the Optineurin LIR by TBK1 can dramatically promote the interaction of Optineurin with FIP200 Claw (Fig. 3e). In contrast, a recent study showed that Optineurin can directly associate with ATG9A to induce mitophagy without the engagement of FIP200[56]. To validate the direct interaction between Optineurin and ATG9A, we constructed an ATG9A(595–839) fragment that includes the entire C-terminal cytosolic tail of ATG9A based on the recently determined human ATG9A structure[57,58]. Using purified ATG9A(595–839) and full-length Optineurin proteins, our biochemical assays clearly demonstrated that there is no direct interaction between the cytosolic tail of ATG9A and Optineurin in vitro (Supplementary Fig. 18a, b). In addition, the leucine zipper region of Optineurin (residues 133–170), which was identified to be responsible for the interaction of Optineurin with ATG9A[56], can directly interact with the active form of Rab8b, a well-proved Optineurin-binding partner[59–61], based on our FPLC-based biochemical assay (Supplementary Fig. 18c). Therefore, it is unlikely that Optineurin can directly recruit ATG9A to initiate the selective autophagy process. Nevertheless, further studies are required to elucidate the detailed relationship between ATG9A, FIP200, and Optineurin in cells.

In this work, we uncovered that in addition to interacting with ATG8 family proteins, the S177-phosphorylated LIR motif of Optineurin can also directly bind to the Claw domain of FIP200 (Figs. 3e and 4). Further structural comparison analyses revealed that the S177-phosphorylated LIR motif of Optineurin utilizes essentially the same key interface residues to interact with FIP200 and the ATG8 family member LC3B (Fig. 4 and Supplementary Fig. 19). Therefore, FIP200 and ATG8 family proteins should be competing in binding to Optineurin, and, similar to that of P62[49], the recruitments of ULK complex and ATG8 family proteins by Optineurin are likely to be mutually exclusive. Since the TBK1-mediated phosphorylation of S177 in the LIR motif of Optineurin can regulate the interactions of Optineurin with both FIP200 and ATG8 family proteins, more functional work is necessary to dissect the individual function contributed by the TBK1-mediated S177 phosphorylation event in these two different types of interactions for the Optineurin-mediated selective autophagy. Interestingly, using qualitative analytical gel filtration chromatography analysis, we also discovered that CCPG1 FIR2 as well as the phosphomimetic S104E mutant of CCPG1 FIR2 can directly interact with all six mammalian ATG8 orthologues (Supplementary Figs. 20 and 21). Further quantitative analyses of the interactions of CCPG1 FIR2 and p-CCPG1 FIR2 with different ATG8 homologs using fluorescence spectroscopy-based assays revealed that CCPG1 FIR2 binds to six ATG8s with distinct binding affinity $K_d$ values, and the phosphorylation of S104 residue in the FIR2 of CCPG1 can increase the interaction between CCPG1 FIR2 and ATG8 family member by ~2–4 fold (Supplementary Fig. 22). Intriguingly, the FIR2 motif of CCPG1 has some resemblance with the atypical LIR motif of NDP52[62]. However, in contrast to that of NDP52[62], CCPG1 FIR2 and p-CCPG1 FIR2 preferentially bound to GABARAP with relatively strong $K_d$ values, ~27 and ~9 µM, respectively (Supplementary Fig. 22). Apparently, due to the potential steric exclusion, FIP200 and ATG8 family members are likely competitive in binding to the FIR motifs of CCPG1. Indeed, further NMR-based titration assays confirmed that there is a competitive interaction between GABARAP and FIP200 Claw for p-CCPG1 FIR2 (Supplementary Fig. 23). However, given that CCPG1 also contains a canonical LIR motif (CGWTVI) N-terminally adjacent to the FIR1 motif, thus further studies are required to elucidate the precise relationship between FIP200 and ATG8 family proteins in binding to CCPG1 in the future.

In this study, based on our biochemical and structural results, we systemically defined the consensus FIR core motif, and revealed that the interaction between FIR motif and FIP200 Claw can be regulated by distinct types of phosphorylation events in the FIR region. Strikingly, the consensus FIR motif uncovered in this study is very similar to some canonical LIR motifs with additional acidic residues immediately preceding the hydrophobic LIR core sequences. Given a large number of LIR-containing proteins in mammalian genomes, it is likely that many currently known LIR motifs but un-characterized in this study may also interact with FIP200 Claw using a similar binding mode as that we have discovered in this study. Indeed, further sequence analyses of the currently known LIR motifs found in mammalian autophagy receptors revealed that many LIR motifs tally with the criteria for a FIP200 Claw-binding FIR defined in this study (Supplementary Table 3). In the future, it will be interesting to know whether they can function as real FIR motifs to recognize FIP200 Claw for the initiation of autophagosome biogenesis during selective autophagy.

Finally, based on our results together with previous peoples' studies[30,49–51], we proposed a model depicting the relationship between FIR/LIR-containing autophagy receptor, ULK complex and ATG8 family proteins during the initiation of autophagosome formation in selective autophagy (Supplementary Fig. 23). In this model, the autophagic cargo, such as ubiquitinated dysfunctional

mitochondria, was firstly recognized by FIR/LIR-containing autophagy receptor, which in turn recruits the FIP200-containing ULK complex mediated by the specific interaction between the FIR/LIR motif of autophagy receptor and the Claw domain of FIP200 (Supplementary Fig. 23). Once, the ULK complex was activated, it would in situ initiate autophagosome formation by promoting the recruitment of downstream autophagic machinery, which subsequently leads to the PE-lipidation and the enrichment of ATG8 family proteins on the nascent preautophagosomal membrane (Supplementary Fig. 23). Then, the enriched PE-conjugated ATG8 family proteins can compete with the FIP200 subunit of ULK complex for binding to the cargo-bound autophagy receptor, and ultimately aid the expansion and closure of the preautophagosomal membrane around the autophagic cargo, forming the autophagosome (Supplementary Fig. 23).

## Methods

**Materials**. Sf9 cell line was kindly provided by Prof. Junying Yuan from Interdisciplinary Research Center on Biology and Chemistry, CAS, Shanghai, China. The full-length human FIP200, Optineurin, TBK1, and Rab8b were obtained from Prof. Jiahuai Han from the School of Life Sciences, Xiamen University, Xiamen, China. The human ATG9 was obtained from Prof. Qiming Sun from the School of Medicine, Zhejiang University, Hangzhou, China. The full-length CCPG1 was synthesized by Sangon Biotech (Shanghai). The synthetic peptide TASDDSDIVTLEPPK (CCPG1 FIR2), TASDDpSDIVTLEPPK (p-CCPG1 FIR2), SSEDSFVEIRMAE (Optineurin LIR), SSEDpSFVEIRMAE (p-Optineurin LIR), where pS corresponds to the phosphorylated serine residue, were purchased from the ChinaPeptides company, and the purities of the commercially synthesized peptides were >98%.

**Protein expression and purification**. The different DNA fragments encoding human FIP200 (residues 1450–1594, 1490–1594), human CCPG1 (residues 99–113), human ATG9 (residues 595–839), human Optineurin (residues 33–209, 33–168, 133–170, 169–185, 186–209) and mutants were cloned into the pET-SUMO vector (a modified version of pET28a vector containing an N-terminal His$_6$-tag and SUMO-tag), the pRSF-Trx vector (a modified version of pRSF vector containing an N-terminal Trx-tag and His$_6$-tag), pET-MBP vector (a modified version of pET32a vector containing an N-terminal His$_6$-tag and MBP-tag), or the pET-32M vector (a modified version of pET32a vector containing an N-terminal Trx-tag and His$_6$-tag) for recombinant protein expressions.

Recombinant proteins were expressed in BL21 (DE3) E. coli cells induced by 100 μM IPTG at 16 °C. The bacterial cell pellets were re-suspended in the binding buffer (50 mM Tris, 500 mM NaCl, 5 mM imidazole at pH 7.9), and then lysed by the ultrahigh-pressure homogenizer FB-110XNANO homogenizer machine (Shanghai Litu Machinery Equipment Engineering Co., Ltd.). Then the lysis was spun down by centrifuge at 35,000×g for 30 min to remove the pellets fractions. His$_6$-tagged proteins were purified by Ni$^{2+}$-NTA agarose (GE Healthcare) affinity chromatography. Each recombinant protein was further purified by size-exclusion chromatography. The N-terminal Trx-tag or SUMO-tag was cleaved by 3C protease or Ulp1, and then further removed by size-exclusion chromatography. Uniformly $^{15}$N-labeled FIP200 fragment and GABARAP proteins were prepared by growing bacteria in M9 minimal medium using $^{15}$NH$_4$Cl (Cambridge Isotope Laboratories Inc.) as the sole nitrogen source.

**Analytical gel filtration chromatography**. Purified proteins were loaded on to a Superose 200 increase 10/300 GL column (GE Healthcare) equilibrated with a buffer containing 20 mM Tris-HCl (pH 7.9), 100 mM NaCl and 1 mM DTT. Analytical gel filtration chromatography was carried out on an AKTA FPLC system (GE Healthcare). The fitting results were further output to the Origin 8.5 software and aligned with each other.

**NMR spectroscopy**. The $^{15}$N-labeled protein samples for NMR titration experiments were concentrated to ~0.2 mM. All the protein samples for NMR studies were prepared in the 50 mM potassium phosphate buffer containing 100 mM NaCl, and 1 mM DTT at pH 6.5, and NMR spectra were acquired at 25 °C on an Agilent 800 MHz spectrometer equipped with an actively z gradient shielded triple resonance cryogenic probe at the Shanghai Institute of Organic Chemistry.

**Analytical ultracentrifugation**. Sedimentation velocity experiments were performed on a Beckman XL-I analytical ultracentrifuge equipped with an eight-cell rotor under 142,250×g at 20 °C. The partial specific volume of different protein samples and the buffer density were calculated using the program SEDNTERP (http://www.rasmb.org/). The final sedimentation velocity data were analyzed and fitted to a continuous sedimentation coefficient distribution model using the program SEDFIT[63]. The fitting results were further output to the Origin 8.5 software and aligned with each other.

**Fluorescence polarization assay**. Fluorescence anisotropy binding assays were performed on the SpectraMax i3x Multi-Mode Detection Platform from Molecular Devices, using a 485 nm excitation filter and a 535 nm emission filter. Peptides were labeled with fluorescein isothiocyanate isomer I (Sigma-Aldrich) at their N-terminals. In this assay, the FITC-labeled peptide (~0.25 μM) was titrated with an increasing amount of testing proteins in a 20 mM Tris (pH 7.9) buffer containing 100 mM NaCl, 1 mM DTT at 25 °C. The $K_d$ values were obtained by fitting the titration curves with the classical one-site binding model using GraphPad Prism 6 software.

**Protein crystallization and structural elucidation**. Crystals of FIP200 (1490–1594), FIP200(1490–1594)/p-CCPG1 FIR2 complex and FIP200 (1490–1594)/p-Optineurin LIR complex were obtained using the sitting-drop vapor-diffusion method at 16 °C. The fresh purified FIP200(1490–1594) protein (10 or 20 mg/ml in 20 mM Tris-HCl, 100 mM NaCl, 1 mM DTT at pH 7.9) was mixed with equal volumes of reservoir solution containing 0.1 M Sodium citrate tribasic dihydrate pH 5.0, 30% (v/v) Jeffamine® ED-2001 pH 7.0. For protein complexes, the fresh purified FIP200(1490–1594) protein was saturated with p-CCPG1 FIR2 peptide or p-Optineurin LIR peptide with a molar ratio up to 1:5, and then adding the freshly mixed proteins (10 or 20 mg/ml in 20 mM Tris-HCl, 100 mM NaCl, 1 mM DTT at pH 7.9) with equal volumes of reservoir solution containing 400 mM KCl, 45% Pentaerythritol ethoxylate (15/4 EO/OH), 100 mM MES for the FIP200(1490–1594)/p-CCPG1 FIR2 complex, or containing 0.2 M Potassium iodide, 0.1 M MES pH 6.5, 25% (w/v) PEG4000 for the FIP200 (1490–1594)/p-Optineurin LIR complex. Before diffraction experiments, glycerol was added as the cryo-protectant. A 1.8 Å resolution X-ray data set for the apo-form FIP200(1490–1594), a 1.4 Å resolution X-ray data set for FIP200(1490–1594)/p-CCPG1 FIR2 complex, and a 2.0 Å resolution X-ray data set for FIP200 (1490–1594)/p-Optineurin LIR complex were collected at the beamline BL17U or BL19U1 of the Shanghai Synchrotron Radiation Facility[64]. The diffraction data were processed and scaled using HKL2000[65].

The phase problem of FIP200(1490–1594), FIP200(1490–1594)/p-CCPG1 FIR2 complex or FIP200(1490–1594)/p-Optineurin LIR complex was solved by molecular replacement method using the FIP200 Claw structure (PDB ID: 6DCE) as the search model with PHASER[66]. The initial structural models were rebuilt manually using COOT[67], and then refined using PHENIX[68]. Further manual model building and adjustments were completed using COOT[67]. The qualities of the final models were validated by MolProbity[69]. The final refinement statistics of solved structures in this study were listed in Supplementary Tables 1 and 2. All the structural diagrams were prepared using the program PyMOL (http://www.pymol.org/).

**In vitro phosphorylation and dephosphorylation assay**. Trx-tagged Optineurin (33–209) purified from E. coil cells and Trx-tagged TBK1(1–674) purified from Sf9 cells were stored in the kinase buffer containing 40 mM Tris, 20 mM MgCl$_2$, 100 μM ATP, and 2 mM DTT at pH 7.5. In vitro phosphorylation assay was performed with 5.3 μM trx-TBK1(1–674) and 106.7 μM trx-Optineurin(33–209) in 1 mL kinase buffer for 30 min at 30 °C, and after the reaction, 50 μL reaction sample was quenched by the addition of 2× sample loading buffer and boiling at 100 °C for 10 min. For λPPase-based dephosphorylation assay, the phosphorylation sample was mixed with manufacturer-provided buffer (10× PMP, 10 mM MnCl$_2$) and lambda protein phosphatase (800 U) in 50 μL total volume for 30 min at 30 °C, and then the reaction was quenched by the addition of 2× sample loading buffer and boiling at 100 °C for 10 min. Finally, the protein samples obtained from the phosphorylation and dephosphorylation assays were analyzed by SDS-PAGE or Phos-tag SDS-PAGE that uses an SDS-PAGE gel containing 50 μM Phos-tag acrylamide with 100 μM MnCl$_2$ for detecting the band shift that represents phosphorylated proteins.

**Mass spectrometry and data analysis**. The phosphorylated protein sample used for the protein identification by LC/tandem MS (MS/MS) was obtained by in vitro phosphorylation of Trx-Optineurin(33–209) by purified Trx-TBK1(1–674). For MS analyses, proteins were precipitated by acetone, and then dissolved in 8 M urea (100 mM Tris-HCl, pH 8.5). TCEP (final concentration is 5 mM) (Thermo Scientific) and iodoacetamide (final concentration is 10 mM) (Sigma) for reduction and alkylation. The protein mixture was digested with Trypsin at 1:50 (w/w) (Promega) overnight. The digestion was stopped by adding 5% formic acid (final concentration) and desalted by mono spin C18 column (GL Science). The peptide mixture was analyzed by a home-made 30 cm-long pulled-tip analytical column (75 μm ID packed with ReproSil-Pur C18-AQ 1.9 μm resin, Dr. Maisch GmbH), the column was then placed in-line with an Easy-nLC 1200 nano HPLC (Thermo Scientific, San Jose, CA) for mass spectrometry analysis. The analytical column temperature was set at 55 °C during the experiments. The mobile phase and elution gradient used for peptide separation were as follows: 0.1% formic acid in water as buffer A and 0.1% formic acid in 80% acetonitrile as buffer B, 0–95 min, 2–35% B; 95–103 min, 35–60% B; 103–104 min, 60–100% B, 104–120 min, 100% B. The flow rate was set as 300 nL/min. Data-dependent tandem mass spectrometry (MS/MS) analysis was performed with a Q Exactive Orbitrap mass spectrometer (Thermo Scientific, San Jose, CA). Peptides eluted from the LC column were directly electrosprayed into the mass spectrometer with the application of a distal 2.5-kV spray

voltage. A cycle of one full-scan MS spectrum ($m/z$ 300–1800) was acquired followed by top 20 MS/MS events, sequentially generated on the first to the twentieth most intense ions selected from the full MS spectrum at a 28% normalized collision energy.

The acquired MS/MS data were analyzed against a UniProtKB Human (database released on Sep. 30, 2018) using Proteome Discoverer 2.4 (Thermo Scientific). Mass tolerances for precursor ions were set at 20 ppm and for MS/MS were set at 0.02 Da. Trypsin was defined as a cleavage enzyme; Cysteine alkylation by iodoacetamide was specified as a fixed modification with mass shift 57.02146. Methionine oxidation and serine/threonine/tyrosine phosphorylation were set as variable modification. In order to accurately estimate peptide probabilities and false discovery rates, we used a decoy database containing the reversed sequences of all the proteins appended to the target database.

**Reporting summary**. Further information on research design is available in the Nature Research Reporting Summary linked to this article.

## Data availability
The coordinates and structure factors of the FIP200(1490–1594), FIP200(1490–1594)/p-CCPG1 FIR2 complex, and FIP200(1490–1594)/p-Optineurin LIR complex mentioned in this study have been deposited in the Protein Data Bank with accession number 7CZG, 7D0E, and 7CZM, respectively. All additional experimental data are available from the corresponding author on request. Source data are provided with this paper.

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

## Acknowledgements
We thank SSRF BL17U and BL19U1 for X-ray beam time, Prof. Jiahuai Han for the full-length FIP200, Optineurin and TBK1 cDNA, and Prof. Qiming Sun for the full-length ATG9A cDNA. This work was supported by grants from the National Key R&D Program of China (2016YFA0501903), the National Natural Science Foundation of China (21822705, 91753113, 21621002, 32071219), the Science and Technology Commission of Shanghai Municipality (20XD1425200, 17JC1405200), the Strategic Priority Research Program of the Chinese Academy of Sciences (XDB20000000), and the start-up fund from State Key Laboratory of Bioorganic and Natural Products Chemistry and Chinese Academy of Sciences.

## Author contributions
Z.Z. and L.P. designed research; Z.Z., J.L., T.F., P.W., Y.W., X.G., M.Z., Y.L., Y.W., M.L., and X.X. performed research; Z.Z., J.L., C.P., and L.P. analyzed data; Z.Z. and L.P. wrote the paper.

## Competing interests
The authors declare no competing interests.
