## [Peer Review File · Nature Communications]

REVIEWER COMMENTS

Reviewer #1 (Remarks to the Author):

In the manuscript by Zhou et al. the authors provide structural insights into the interaction of autophagy cargo receptors with the C-terminal Claw domain of FIP200.

The FIP200 scaffold protein is a subunit of the ULK1 kinase complex and is important for the initiation of autophagosome formation around specific cellular cargoes. In recent years it has become clear that autophagy receptors including p62, NDP52 and TAX1BP1 initiate autophagosome biogenesis in vicinity of the cargo by recruiting the FIP200 protein. The C-terminal Claw domain, for which the structure was determined recently (PMID 30853400), was shown to bind to a peptide of p62 but a co-crystal structure of the Claw-receptor complex has not yet been reported.

Here the authors conduct a thorough structural and biochemical analysis of the interaction of the Claw domain of human FIP200 with the FIP200 binding regions (FIRs) of the receptors CCPG1 and OPTN. They find that these interactions are positively regulated by phosphorylation and that their FIRs and their ATG8 binding sites overlap, suggesting mutual exclusive interactions, as shown before for p62 (PMID 30853400). In addition, a large structural rearrangement in the Claw dimer is revealed upon binding of CCPG1. The consequences of this conformational change remain unexplored.

Overall, this is a carefully conducted study but its conceptual advance is limited, making it interesting for the selective autophagy community but not for a wider audience. Below are a few comments, mainly with respect to claims by the authors, that are overstatements in this reviewer's opinion because related findings were already reported in the existing literature.

1. The authors claim in the abstract that "the mechanistic basis underlying the specific interactions of autophagy receptors with FIP200 and the relevant regulatory mechanism remain elusive" and further in the introduction "However, the detailed molecular mechanism governing the recruitment of upstream autophagy machinery for autophagosome biogenesis by autophagy receptors as well as the relevant phosphorylation-dependent regulatory mechanism during selective autophagy remains elusive.". These are overstatements. While it is correct that many details of this interactions are unclear, biochemical and structural insights into these binding events are available (PMIDs: 30853400, 32773036, 30853402).
2. The authors write "In addition, further analytical ultracentrifugation-based assay revealed that FIP200 Claw forms a stable homo-dimer". This was already shown in PMID: 30853400
3. It is stated that "therefore some autophagy receptors might also directly interact with FIP200 Claw through their LIR regions and regulated by relevant kinase-mediated phosphorylation." This was already shown in PMID: 30853400
4. The authors state „how FIP200 communicates with other autophagic proteins are still largely unknown". However, there are several recent publications providing insights into these interactions. PMIDs: 30853400, 32773036, 30853402, 30853401, 29290589, 32516362
5. Can the authors provide evidence for the phosphorylation of S104 located in the CCPG1 FIR2 in cells?
6. In Figure 2 or the corresponding supplemental figure, it would be helpful to show the claw and the bound receptor peptide in context of the preceding coiled coil domain.
7. The order of sub-panels in Figure 3 is odd and hard to follow.

Reviewer #2 (Remarks to the Author):

Zhou et al. report the structure of the FIP200 Claw domain bound to phosphorylated FIR peptides from OPTN and CCPG1. Previously, the structure of the apo-Claw had been determined, and solving structures of cargo-peptide bound complexes was the next natural step in this field. The structures are at high to very high resolution and of high quality. The crystallographic data reporting and deposition is appropriate. They will definitely be interesting to specialists in autophagy structural biology. The structures are corroborated by high quality fluorescence polarization binding assays that cover a two order of magnitude range of affinities from the best phospho-CCPG1 peptide to the most impaired mutant. NMR and sedimentation data are also provided.

The weaknesses are in the lack of biological data, which could perhaps be forgiven considering the value of the structures, but the interpretation is confusing relative to the literature in one case, and unclear in the other. CCPG1 was speculated to undergo phosphoregulation by Smith et al & Wilkinson, *Dev Cell* (2018), however, there are still no biological data available, so the significance remains quite speculative. Identification of the kinase and demonstration of a functional role in ER-phagy would be a major advance but understandably is beyond the scope of this manuscript. OPTN is well known to be phosphoregulated by TBK1, and it is quite interesting that the affinity increase is so substantial, however, Yamano et al. *J Cell Biol* (2020) just published that OPTN does not need to engage with FIP200 to induce mitophagy, but rather acts through ATG9. Seeing the new results from Zhou makes me wonder if Yamano et al. have missed something. Nevertheless, some clarification would be needed of this discrepancy.

Reviewer #3 (Remarks to the Author):

In this manuscript, Zhou et al describe biophysical and biochemical characterization of the complex formed by the interaction between the Claw domain of the mammalian core autophagy protein FIP200 (also known as RB1CC1) and the cognate FIP200-Interacting Region (FIRs motifs of two selective autophagy receptors (SARs), CCPG1 (SAR for ERphagy) and Optineurin (SAR for aggrephagy), respectively. Selective autophagy is a field of biology key for understanding of the pathogenesis of a number of human diseases, such as neurodegeneration and infection diseases. Therefore, mechanistic understanding of this pathway is of great importance for developing new therapeutics.

By performing crystallography and additional analytical methods, such as gel filtration chromatography, ultracentrifugation and fluorescence polarization (FP) assay, the authors uncover molecular features of the FIP200:FIR interaction, such as the dimeric nature of the Claw domain and its ability to establish hydrophobic and polar contacts to the FIP200-Interacting Regions (FIRs) of the SARs. The authors explain the importance of upstream, negatively charged amino acid residues, including the phosphorylatable Ser residues in FIR2 of CCPG1 and the novel FIR in Optineurin, which drive the affinity of the Claw:FIR interaction.

Perhaps, the most striking and impactful conclusion of the authors is their suggestion, based on the results obtained in their present study, that the well-accepted LC3-Interacting region (LIR) concept (key for the function of multiple SARs) requires revision and needs to be expanded to include also the FIR consensus sequence. This observation is backed up by the previous publication on the FIR/LIR overlap in p62/SQSTM1 (Turco et al 2019 *Mol Cell*). The previously and herein suggested steric clash of LC3/GABARAP with FIP200 for binding to the SARs raises important questions regarding the mechanics of the interaction between the SARs and the core autophagy machinery critical for the cargo-induced formation of the selective autophagosome. The technical execution of the study is of

very high quality, so that the conclusions are very well founded. I have only few suggestions on how to further improve the manuscript, given the importance of the discussed molecular context, in order to secure publication in Nature Communications.

Specific points:

1. The authors should discuss their findings in reference to the solved structure of the FIP200 Claw domain published previously by Turco et al (2019 Mol Cell). What are similarities and differences of the two dimeric structures? How does the combined FIR/LIR in p62/SQSTM1 with the published phosphoregulatable sites relate to the FIR/LIR in CCPG1 and Optineurin? These are important points that can and should be mentioned in this type of work to gain significant advancement over the previously published work.
2. Another important aspect is the direct comparison of the Claw:FIR vs. LC3/GABARAP:LIR interactions. The authors do mention this concept in their manuscript and also provide some data to prove that LC3/GABARAPs interact with CCPG1 FIR motifs (e.g., Suppl Fig 17-19). Of note, CCPG1 also has a canonical LIR motif that interacts with LC3/GABARAPs (Smith et al 2018 Dev Cell). It would be of great importance and also within the reach of this team to test competitive interaction between a member of the LC3/GABARAP family (e.g., GABARAP) and the Claw domain of FIP200 for a FIR/LIR. Can the authors provide experimental evidence (a simple titration experiment) for their suggestion that LC3/GABARAPs would compete for the same binding motif?
3. It would also be very useful to provide in this manuscript, based on the two available structures of the FIR:Claw complexes and the known LIR:LC3 structures, a few models of the predicted FIR/LIRs (e.g. p62/SQSTM1, NBR1 and FAM134b) in complex with the Claw domain of FIP200. This would further strengthen the impact of their findings on the field of selective autophagy.
4. On the other hand, ATG16L1 FIR was not described to bind LC3/GABARAP directly, which clearly sets this FIR apart from those in dual FIR/LIRs in SARs. This should also be discussed in the manuscript.
5. Given the scope of the paper, I am not requesting any experiments in cellular selective autophagy assays for testing the functional consequences of disrupting LIR/FIR or dimerization of the Claw domain. However, it would be appropriate to comment on anticipated effects of FIR/LIR on aggregatephagy, xenophagy, and mitophagy. Here, a figure describing the model on the relationship between FIP200, LC3/GABARAP and the SARs in selective autophagosome formation would be a welcome illustration of the central idea of this otherwise great manuscript.

Point-by-point responses to the reviewers' comments:

(Reviewers' comments are in **blue**, and our responses are in **black**)

Specific responses to the criticisms from the Reviewer #1:

We appreciate review's constructive criticisms and insightful suggestions to our manuscript, as these have been valuable in improving the scientific quality of the manuscript. The following is our detailed responses to the reviewer's comments:

In the manuscript by Zhou et al. the authors provide structural insights into the interaction of autophagy cargo receptors with the C-terminal Claw domain of FIP200.

The FIP200 scaffold protein is a subunit of the ULK1 kinase complex and is important for the initiation of autophagosome formation around specific cellular cargoes. In recent years it has become clear that autophagy receptors including p62, NDP52 and TAX1BP1 initiate autophagosome biogenesis in vicinity of the cargo by recruiting the FIP200 protein. The C-terminal Claw domain, for which the structure was determined recently (PMID 30853400), was shown to bind to a peptide of p62 but a co-crystal structure of the Claw-receptor complex has not yet been reported.

Here the authors conduct a thorough structural and biochemical analysis of the interaction of the Claw domain of human FIP200 with the FIP200 binding regions (FIRs) of the receptors CCPG1 and OPTN. They find that these interactions are positively regulated by phosphorylation and that their FIRs and their ATG8 binding sites overlap, suggesting mutual exclusive interactions, as shown before for p62 (PMID 30853400). In addition, a large structural rearrangement in the Claw dimer is revealed upon binding of CCPG1. The consequences of this conformational change remain unexplored.

Overall, this is a carefully conducted study but its conceptual advance is limited, making it interesting for the selective autophagy community but not for a wider audience. Below are a few comments, mainly with respect to claims by the authors, that are overstatements in this reviewer's opinion because related findings were already reported in the existing literature.

1. The authors claim in the abstract that "the mechanistic basis underlying the specific interactions of autophagy receptors with FIP200 and the relevant regulatory mechanism remain elusive" and further in the introduction "However, the detailed molecular mechanism governing the recruitment of upstream autophagy machinery for autophagosome biogenesis by autophagy receptors as well as the relevant phosphorylation-dependent regulatory mechanism during selective autophagy remains elusive.". These are overstatements. While it is correct that many details of this interactions are unclear, biochemical and structural insights into these binding events are available (PMIDs: 30853400, 32773036, 30853402).

We thank the reviewer for pointing out these inappropriate statements in our original manuscript. We are sorry for our careless reading and survey of the previously published papers. Following reviewer's comment, we have modified and tuned down our

statements in the revised manuscript. Furthermore, we have taken extra care to improve the writing of our manuscript.

2. The authors write “In addition, further analytical ultracentrifugation-based assay revealed that FIP200 Claw forms a stable homo-dimer”. This was already shown in PMID: 30853400.

Following reviewer’s comment, we have modified our statement to “In line with a previous report¹, further analytical ultracentrifugation-based assay revealed that FIP200 Claw forms a stable homo-dimer, which can simultaneously interact with two monomeric phosphomimetic S104E mutants of CCPG1 FIR2 to form a hetero-tetramer in solution.” in the revised manuscript.

3. It is stated that “therefore some autophagy receptors might also directly interact with FIP200 Claw through their LIR regions and regulated by relevant kinase-mediated phosphorylation.” This was already shown in PMID: 30853400.

We thank the reviewer for pointing out this mistake in our writing. According to reviewer’s comment, we have modified our statement to “Notably, the LIR region of P62 was recently demonstrated to participate in the interaction with FIP200 Claw, and this interaction can be further enhanced by the phosphorylation of four residues located in the LIR region of P62¹. Therefore, we inferred that other autophagy receptors might also directly interact with FIP200 Claw through their LIR regions and regulated by relevant kinase-mediated phosphorylation.” in the revised manuscript.

4. The authors state “how FIP200 communicates with other autophagic proteins are still largely unknown”. However, there are several recent publications providing insights into these interactions. PMIDs: 30853400, 32773036, 30853402, 30853401, 29290589, 32516362.

We thank the reviewer for pointing out this inappropriate statement in our original manuscript. Following reviewer’s comment, we have tuned down our statement and removed this sentence in the revised manuscript.

5. Can the authors provide evidence for the phosphorylation of S104 located in the CCPG1 FIR2 in cells?

We thank the reviewer for the above constructive comment and encouragement. Actually, we are also very interested in the speculative phosphorylation event of CCPG1, which was suggested by a previous study from Simon Wilkinson’s group². However, given that CCPG1 is an ER-resident transmembrane protein and due to our technical limitation, we are unable to perform cellular experiments to evaluate the phosphorylation status of CCPG1 S104 in cells. Meanwhile, unfortunately, we also can’t find a good collaborator to help us to test this type of assay in a limited time for revision. Thus, we prefer to pursue the cellular phosphorylation event of CCPG1 as well as the related upstream kinase in a follow-up study.

6. In Figure 2 or the corresponding supplemental figure, it would be helpful to show the claw and the bound receptor peptide in context of the preceding coiled coil domain.

Following reviewer’s suggestion, we have added a supplemental figure to show the structural comparisons of FIP200 Claw/p-CCPG1 FIR2 complex with the determined

apo-form structure of a FIP200 C-terminal fragment including the Claw domain and the preceding coiled-coil domain (PDB ID: 6GMA) (see **Fig. I** below, and **Supplemental Fig. 9** in the revised manuscript). Particularly, in panel **a** of this supplemental figure, we overlay these two dimeric structures by aligning selected one monomeric Claw domain in these two structures, while in panel **b**, the two monomeric FIP200 C-terminal fragments are individually aligned to the two Claw domains in the FIP200 Claw/p-CCPG1 FIR2 complex. Based on these structural analyses, it is apparently that the rearrangement of the FIP200 Claw dimer induced by p-CCPG1 FIR2 binding may cause large conformational changes of the coiled-coil domain preceding the Claw of FIP200. We have included these new structural analyses in the revised manuscript.

Figure I: The comparisons of the FIP200 Claw/p-CCPG1 FIR2 complex structure with the determined structure of a FIP200 C-terminal fragment including the Claw domain and the preceding coiled-coil domain. (a and b) Ribbon representations showing the structural comparisons of the FIP200 Claw/p-CCPG1 FIR2 complex (slate/orange) with the apo-form FIP200 fragment containing the Claw domain and the preceding coiled-coil domain (olive, PDB ID: 6GMA). In panel **a, the two dimeric structures are overlaid by aligning selected one Claw monomer in these two structures. In panel **b**, the two monomeric FIP200 C-terminal fragments are individually aligned to the two Claw domains of the FIP200 Claw/p-CCPG1 FIR2 complex.**

7. The order of sub-panels in Figure 3 is odd and hard to follow.

We thank the reviewer for pointing out our inappropriate organization of the Figure 3 in the original manuscript. According to reviewer's comment, we have modified and re-

ordered the sub-panels of Figure 3 in the revised manuscript (see **Fig. II** below, and **Fig. 3** in the revised manuscript).

Figure II: TBK1-mediated phosphorylation of the LIR region of Optineurin can promote the interaction between Optineurin and FIP200 Claw. (a) *In-vitro* phosphorylation assays showing that Optineurin(33-209) could be specifically phosphorylated by the purified TBK1 kinase. The gel in the left panel is normal SDS-PAGE gel, while the right one is a phos-tag gel. (b and c) Analytic gel filtration chromatography-based analyses of the interactions of FIP200 Claw domain with Optineurin(33-209) (b), and the phosphorylated Optineurin(33-209) mediated by TBK1, which is named as p-Optineurin(33-209) (c). (d) The figure shows a higher-energy collisional dissociation (HCD) MS/MS spectrum recorded on the $[M+2H]^{2+}$ ion at m/z 853.87 of the human Optineurin peptide LNSSGSSEDSFVEIR harboring one phosphorylated site (denoted by lowercase s). Predicted b- and y-type ions (not including all) are listed above and below the peptide sequence, respectively. Ions observed are labeled in the spectrum and indicate that the S177 residue of Optineurin(33-209) protein is modified with the phosphate. (e) Fluorescence polarization assays measure the binding affinities of FIP200 Claw domain (residues 1490-1594) with Optineurin LIR (residues 169-185) (black) and the phosphorylated Optineurin LIR (p-Optineurin LIR) (red). K_d values are the fitted dissociation constants with standard errors, when using the one-site binding model to fit the FP data.

Specific responses to the criticisms from the Reviewer #2:

We are very grateful to the strong supports and constructive suggestions from the reviewer. The following is our detailed responses to the reviewer's comments:

Zhou et al. report the structure of the FIP200 Claw domain bound to phosphorylated FIR peptides from OPTN and CCPG1. Previously, the structure of the apo-Claw had been

determined, and solving structures of cargo-peptide bound complexes was the next natural step in this field. The structures are at high to very high resolution and of high quality. The crystallographic data reporting and deposition is appropriate. They will definitely be interesting to specialists in autophagy structural biology. The structures are corroborated by high quality fluorescence polarization binding assays that cover a two order of magnitude range of affinities from the best phospho-CCPG1 peptide to the most impaired mutant. NMR and sedimentation data are also provided.

The weaknesses are in the lack of biological data, which could perhaps be forgiven considering the value of the structures, but the interpretation is confusing relative to the literature in one case, and unclear in the other. CCPG1 was speculated to undergo phosphoregulation by Smith et al & Wilkinson, *Dev Cell* (2018), however, there are still no biological data available, so the significance remains quite speculative. Identification of the kinase and demonstration of a functional role in ER-phagy would be a major advance but understandably is beyond the scope of this manuscript. OPTN is well known to be phosphoregulated by TBK1, and it is quite interesting that the affinity increase is so substantial, however, Yamano et al. *J Cell Biol* (2020) just published that OPTN does not need to engage with FIP200 to induce mitophagy, but rather acts through ATG9. Seeing the new results from Zhou makes me wonder if Yamano et al. have missed something. Nevertheless, some clarification would be needed of this discrepancy.

We understand reviewer's concern, and we also seek to investigate the cellular phosphorylation event of CCPG1 as well as the related upstream kinase. However, due to our technical limitation, we can't figure out these issues in a limited time for revision. We will address this speculative phosphorylation event of CCPG1 and the relevant kinase in a follow-up paper. As for a direct interaction between ATG9 and Optineurin proposed by a recent study from Yamano et al³, we are also very interested in this finding. In their study, they used a phase-separated fluorescent foci system, which was artificially generated by a homooligomeric Ash tag fused with a linear 6xUb chain and a homotetrameric humanized Azami-Green (hAG) tag fused with Optineurin (actually our previous biochemical and structural studies well demonstrated that Optineurin forms a stable dimer not a tetramer in solution^{4,5}), and a Co-IP assay to show that Optineurin can directly associate with ATG9A in cells³. However, those assays can't exclude the possibility that Optineurin can indirectly interact with ATG9A mediated by other unknown adaptor proteins or specific proteins resided on the ATG9A-containing vesicles, which can interact with Optineurin. To clarify reviewer's concern and to validate the potential direct interaction between Optineurin and ATG9A, we constructed a ATG9A(595-839) fragment that includes the entire C-terminal cytosolic tail of ATG9A based on the recently determined human ATG9A structure^{6,7}. Using purified ATG9A(595-839) and full length Optineurin proteins, our biochemical assays clearly demonstrated that there is no direct interaction between ATG9A and Optineurin *in vitro* (see **Fig. IIIa, b** below, and **Supplementary Fig. 18a, b** in the revised manuscript). In addition, the leucine zipper region of Optineurin (residues 133-170), which was identified to be responsible for the association of Optineurin with ATG9A³, can directly interact with the active form of Rab8b, a well-proved Optineurin-binding partner⁸⁻¹⁰, based on our FPLC-based biochemical assay (see **Fig. IIIc** below, and **Supplementary Fig. 18c** in the revised manuscript, Rab8b Q67L mutant is a constitutively active mutant of Rab8b).

Meanwhile, the reason why we tested the interaction between FIP200 and Optineurin in our manuscript is that we found FIP200 is one of top candidates for interacting with Optineurin when searching for unknown cellular binding partners of Optineurin using a TAP/MS (tandem affinity purification coupled with MS analysis) based assay conducted in 2016 (data not shown). Consistently, a previous elegant study from Richard J. Youle’s group clearly demonstrated that the function of Optineurin in Parkin-mediated mitophagy involves the recruitment of the ULK complex¹¹. Therefore, it is unlikely that Optineurin can directly recruit ATG9A to initiate selective autophagy process. Nevertheless, further studies are required to elucidate the detailed relationship between ATG9A and Optineurin. We have discussed this part in the revised manuscript.

Figure III: Biochemical characterizations of the interactions of Optineurin with ATG9A and Rab8b. (a) Analytic gel filtration chromatography-based analyses of the interaction of MBP-tagged ATG9A(595-839) with full-length Optineurin. (b) The SDS-PAGE combined with Coomassie-blue staining analysis of the protein components of the input MBP-tagged ATG9A(595-839) protein as well as the indicated “fraction 1” fractions collected from the analytic gel filtration chromatography experiments of Trx-

tagged Optineurin (the blue curve) and the MBP-ATG9A(595-839)/Trx-Optineurin mixture (the black curve). (c) Analytic gel filtration chromatography-based analysis of the interaction between the constitutively active Rab8b Q67L mutant and the Optineurin(133-170) fragment.

Specific responses to the criticisms from the Reviewer #3:

We thank the reviewer for the strong supports and enthusiastic comments of our manuscript. The following is a list of our detailed responses to the comments raised:

In this manuscript, Zhou et al describe biophysical and biochemical characterization of the complex formed by the interaction between the Claw domain of the mammalian core autophagy protein FIP200 (also known as RB1CC1) and the cognate FIP200-Interacting Region (FIRs motifs of two selective autophagy receptors (SARs), CCPG1 (SAR for ERphagy) and Optineurin (SAR for aggrephagy), respectively. Selective autophagy is a field of biology key for understanding of the pathogenesis of a number of human diseases, such as neurodegeneration and infection diseases. Therefore, mechanistic understanding of this pathway is of great importance for developing new therapeutics.

By performing crystallography and additional analytical methods, such as gel filtration chromatography, ultracentrifugation and fluorescence polarization (FP) assay, the authors uncover molecular features of the FIP200:FIR interaction, such as the dimeric nature of the Claw domain and its ability to establish hydrophobic and polar contacts to the FIP200-Interacting Regions (FIRs) of the SARs. The authors explain the importance of upstream, negatively charged amino acid residues, including the phosphorylatable Ser residues in FIR2 of CCPG1 and the novel FIR in Optineurin, which drive the affinity of the Claw:FIR interaction.

Perhaps, the most striking and impactful conclusion of the authors is their suggestion, based on the results obtained in their present study, that the well-accepted LC3-Interacting region (LIR) concept (key for the function of multiple SARs) requires revision and needs to be expanded to include also the FIR consensus sequence. This observation is backed up by the previous publication on the FIR/LIR overlap in p62/SQSTM1 (Turco et al 2019 Mol Cell). The previously and herein suggested steric clash of LC3/GABARAP with FIP200 for binding to the SARs raises important questions regarding the mechanics of the interaction between the SARs and the core autophagy machinery critical for the cargo-induced formation of the selective autophagosome. The technical execution of the study is of very high quality, so that the conclusions are very well founded. I have only few suggestions on how to further improve the manuscript, given the importance of the discussed molecular context, in order to secure publication in Nature Communications.

Specific points:

1. The authors should discuss their findings in reference to the solved structure of the FIP200 Claw domain published previously by Turco et al (2019 Mol Cell). What are similarities and differences of the two dimeric structures? How does the combined FIR/LIR in p62/SQSTM1 with the published phosphoreglatable sites relate to the

FIR/LIR in CCPG1 and Optineurin? These are important points that can and should be mentioned in this type of work to gain significant advancement over the previously published work.

We thank the reviewer for these constructive comments. Following reviewer's suggestion, we have added more discussion in the revised manuscript related to the comparison of our determined structures of FIP200 Claw/p-CCPG1 FIR2 and FIP200 Claw/p-Optineurin LIR complexes with the previously solved apo-form structures of the FIP200 Claw domain (PDB ID: 6DCE) and the FIP200 C-terminal region containing the Claw domain and the preceding coiled-coil domain (PDB ID: 6GMA) published by Turco et al¹. Also see our detailed response to the point #6 from the Reviewer 1. In addition, we have also discussed the published phosphoregulatable sites of P62 relate to the FIR/LIR of CCPG1 and Optineurin in the revised manuscript.

2. Another important aspect is the direct comparison of the Claw:FIR vs. LC3/GABARAP:LIR interactions. The authors do mention this concept in their manuscript and also provide some data to prove that LC3/GABARAPs interact with CCPG1 FIR motifs (e.g., Suppl Fig 17-19). Of note, CCPG1 also has a canonical LIR motif that interacts with LC3/GABARAPs (Smith et al 2018 Dev Cell). It would be of great importance and also within the reach of this team to test competitive interaction between a member of the LC3/GABARAP family (e.g., GABARAP) and the Claw domain of FIP200 for a FIR/LIR. Can the authors provide experimental evidence (a simple titration experiment) for their suggestion that LC3/GABARAPs would compete for the same binding motif?

Following reviewer's comments, we have purified the ¹⁵N-labelled GABARAP, and conducted a NMR titration-based assay to analysis the competitive interaction between GABARAP and FIP200 Claw domain for p-CCPG1 FIR2. In this assay, ¹⁵N-labelled GABARAP was firstly titrated with un-labelled p-CCPG1 FIR2 peptide at the molar ratio of 1:1. The obtained NMR result showed that many peaks in the ¹H-¹⁵N HSQC spectrum of GABARAP undergo significant chemical shift changes or peak-broadenings (see **Fig. IV** below, and **Supplementary Fig. 23** in the revised manuscript), indicating a direct interaction between GABARAP and p-CCPG1 FIR2. Then, purified FIP200 Claw was further added to this GABARAP/p-CCPG1 FIR2 mixture at the molar ratio of 1:1 or 2:1. As expected, most peaks of GABARAP in the GABARAP/p-CCPG1 FIR2 mixture return back to the original position of the apo-form GABARAP, confirming that FIP200 Claw and GABARAP are competitive in binding to p-CCPG1 FIR2. We have included this new data in the revised manuscript.

Figure IV: NMR-based characterizations of the competitive interaction between GABARAP and FIP200 Claw for p-CCPG1 FIR2. Superposition plots of the ^1H - ^{15}N HSQC spectra of GABARAP (red) titrated with un-labelled p-CCPG1 FIR2 peptide at the molar ratio of 1:1 (green), and then added FIP200 Claw to the mixture at the molar ratio of 1:1 (cyan), or 2:1 (blue). For clarity, the insert shows the enlarged view of a selected region of the overlaid ^1H - ^{15}N HSQC spectra. This NMR titration result clearly demonstrated that FIP200 Claw and GABARAP are competitive in binding to p-CCPG1 FIR2.

3. It would also be very useful to provide in this manuscript, based on the two available structures of the FIR:Claw complexes and the known LIR:LC3 structures, a few models of the predicted FIR/LIRs (e.g. p62/SQSTM1, NBR1 and FAM134b) in complex with the Claw domain of FIP200. This would further strengthen the impact of their findings on the field of selective autophagy.

Following reviewer's comments, we have built three models of the LIR regions of P62, NBR1 and FAM134b in complex with the FIP200 Claw domain based on our determined structures of the FIP200 Claw/p-CCPG1 FIR2 complex and the FIP200 Claw/p-Optineurin LIR complex (see **Fig. V** below, and **Supplementary Fig. 17** in the revised manuscript). In these structural models, the LIR regions of P62, NBR1 and FAM134b can well interact with the FIP200 Claw as that of p-CCPG1 FIR2 or p-Optineurin LIR. We have included these structural models in the revised manuscript.

Figure V: Structural modeling analyses of the potential interactions between FIP200 Claw and the LIR regions of P62, NBR1 and FAM134B. (a-c) The combined surface representation and the ribbon-stick model showing the detailed binding surface between FIP200 Claw and P62 LIR (a), NBR1 LIR (b), or FAM134B LIR (c) in a structural model of the FIP200 Claw/P62 LIR complex, the FIP200 Claw/NBR1 LIR complex, or the FIP200 Claw/FAM134B LIR complex. In these drawings, the LIR regions of P62, NBR1 and FAM134B are displayed in the ribbon-stick model, and the FIP200 Claw domains are showed in the surface representation colored by amino acid types. Specifically, the hydrophobic amino acid residues in the surface model of FIP200 Claw are drawn in yellow, the positively charged residues in blue, the negatively charged residues in red, and the uncharged polar residues in gray. These structural models were initially generated based on the determined crystal structure of the FIP200 Claw/p-CCPG1 FIR2 complex or the FIP200 Claw/p-Optineurin LIR complex using PyMOL (<http://www.pymol.org/>), and were further refined using the YASARA energy minimization server to increase the model accuracy¹².

4. On the other hand, ATG16L1 FIR was not described to bind LC3/GABARAP directly, which clearly sets this FIR apart from those in dual FIR/LIRs in SARs. This should also be discussed in the manuscript.

We understand reviewer's concern. Actually, in addition to its conventional role to associate with ATG5 forming the E3-like ATG16L1/ATG5/ATG12 complex for catalyzing the PE-conjugation of ATG8 family proteins, ATG16L1 also can function as autophagy adaptor in antibacterial xenophagy. Honestly speaking, we have already demonstrated that ATG16L1 FIR can directly interact with six ATG8 family proteins (see **Fig. VIa-f** below). Meanwhile, we have also determined the structures of ATG16L1 FIR in complex with GABARAPL1 and FIP200 Claw (see **Fig. VIg** and **h** below). Therefore, we think ATG16L1 can be classified as a dual FIR/LIR-containing selective autophagy receptor (SAR). Furthermore, we also determined the ATG16L1/WIPI2b complex structure (see **Fig. VIi** below). Given that there are too many biochemical and structural results, we prefer to present those ATG16L1-related data in a follow-up manuscript just focusing on ATG16L1. Therefore, we don't want to include any result and discussion related to ATG16L1 in this manuscript.

Figure VI: Our unpublished biochemical and structural results related to the interactions of ATG16L1 with ATG8 family proteins, FIP200 Claw and WIPI2b. (a-

f) Quantitative ITC-based measurements of the binding affinities of ATG16L1 FIR (residues 235-247) with LC3A (a), LC3B (b), LC3C (c), GABARAP (d), GABARAPL1 (e) and GABARAPL2 (f). K_d values are the fitted dissociation constants with standard errors, when using the one-site binding model to fit the ITC data. These ITC results clearly demonstrate that ATG16L1 FIR can directly bind to different ATG8 proteins. (g) Ribbon diagram showing the overall structure of GABARAPL1 and ATG16L1 FIR complex. In this drawing, GABARAPL1 is shown in green, and ATG16L1 FIR in orange. (h) Ribbon diagram showing the overall structure of the dimeric FIP200 Claw/ATG16L1 FIR complex. In this drawing, FIP200 Claw is shown in blue, and ATG16L1 FIR in orange. (i) Ribbon diagram showing the overall structure of WIPI2b in complex with ATG16L1(207-230) fragment. In this drawing, WIPI2b is shown in blue, and ATG16L1 ATG16L1(207-230) fragment in pink.

5. Given the scope of the paper, I am not requesting any experiments in cellular selective autophagy assays for testing the functional consequences of disrupting LIR/FIR or dimerization of the Claw domain. However, it would be appropriate to comment on anticipated effects of FIR/LIR on aggrephagy, xenophagy, and mitophagy. Here, a figure describing the model on the relationship between FIP200, LC3/GABARAP and the SARs in selective autophagosome formation would be a welcome illustration of the central idea of this otherwise great manuscript.

We thank the reviewer for this nice comment. Following reviewer's suggestion, we have added a cartoon model in the Discussion section of the revised manuscript to describe the relationship between FIR/LIR-containing autophagy receptor, ULK complex and ATG8 family proteins during the initiation of autophagosome formation in selective autophagy (see **Fig. VII** below, and **Supplementary Fig. 24** in the revised manuscript). In this model, the autophagic cargo, such as ubiquitinated dysfunctional mitochondria, was firstly recognized by FIR/LIR-containing autophagy receptor, which in turn recruits the FIP200-containing ULK complex mediated by the specific interaction between the FIR/LIR motif of autophagy receptor and the Claw domain of FIP200. Once, the ULK complex was activated, it would *in situ* initiate autophagosome formation by promoting the recruitment of downstream autophagic machinery, which subsequently leads to the PE-lipidation and the enrichment of ATG8 family proteins on the nascent preautophagosomal membrane. Then, the enriched PE-conjugated ATG8 family proteins can compete with the FIP200 subunit of ULK complex for binding to the cargo-bound autophagy receptor, and ultimately aid the expansion and closure of the preautophagosomal membrane around the autophagic cargo to form the autophagosome.

Figure VII: A schematic cartoon diagram describing the relationship between FIR/LIR-containing autophagy receptor, ULK complex and ATG8 family proteins during the initiation of autophagosome formation in selective autophagy. In this model, LC3 stands for ATG8 family proteins.

References

1. Turco, E. et al. FIP200 Claw Domain Binding to p62 Promotes Autophagosome Formation at Ubiquitin Condensates. *Molecular Cell* **74**, 330-+ (2019).
2. Smith, M.D. et al. CCPG1 Is a Non-canonical Autophagy Cargo Receptor Essential for ER-Phagy and Pancreatic ER Proteostasis. *Developmental Cell* **44**, 217-+ (2018).
3. Yamano, K. et al. Critical role of mitochondrial ubiquitination and the OPTN-ATG9A axis in mitophagy. *J Cell Biol* **219**(2020).
4. Li, F.X. et al. Structural insights into the interaction and disease mechanism of neurodegenerative disease-associated optineurin and TBK1 proteins. *Nature Communications* **7**(2016).
5. Li, F.X. et al. Structural insights into the ubiquitin recognition by OPTN (optineurin) and its regulation by TBK1-mediated phosphorylation. *Autophagy* **14**, 66-79 (2018).
6. Guardia, C.M. et al. Structure of Human ATG9A, the Only Transmembrane Protein of the Core Autophagy Machinery. *Cell Rep* **31**, 107837 (2020).
7. Matoba, K. et al. Atg9 is a lipid scramblase that mediates autophagosomal membrane expansion. *Nature Structural & Molecular Biology* **27**, 1185-U224 (2020).
8. Rezaie, T. et al. Adult-onset primary open-angle glaucoma caused by mutations in optineurin. *Science* **295**, 1077-1079 (2002).
9. De Marco, N., Buono, M., Troise, F. & Diez-Roux, G. Optineurin increases cell survival and translocates to the nucleus in a Rab8-dependent manner upon an apoptotic stimulus. *Journal of Biological Chemistry* **281**, 16147-16156 (2006).

10. Vaibhava, V. et al. Optineurin mediates a negative regulation of Rab8 by the GTPase-activating protein TBC1D17. *J Cell Sci* **125**, 5026-39 (2012).
11. Lazarou, M. et al. The ubiquitin kinase PINK1 recruits autophagy receptors to induce mitophagy. *Nature* **524**, 309-14 (2015).
12. Krieger, E. et al. Improving physical realism, stereochemistry, and side-chain accuracy in homology modeling: Four approaches that performed well in CASP8. *Proteins-Structure Function and Bioinformatics* **77**, 114-122 (2009).

REVIEWERS' COMMENTS

Reviewer #1 (Remarks to the Author):

The authors have addressed all my points and I have no further comments. With the additional data, the manuscript is much stronger.

Reviewer #2 (Remarks to the Author):

The authors have adequately addressed my concerns.

Reviewer #3 (Remarks to the Author):

I am satisfied with the authors' responses and thank them for the very interesting work.

Point-by-point responses to the comments raised by the referees
(Referees' comments are in **blue**, and our responses are in **black**)

Specific responses to the comments from the Reviewer #1:

The authors have addressed all my points and I have no further comments. With the additional data, the manuscript is much stronger.

We thank the reviewer for his/her wonderful supports.

Specific responses to the comments from the Reviewer #2:

The authors have adequately addressed my concerns.

We thank the reviewer for his/her strong supports of our manuscript.

Specific responses to the comments from the Reviewer #3:

I am satisfied with the authors' responses and thank them for the very interesting work.

We appreciate reviewer's constructive suggestions, and thank reviewer's strong supports of our manuscript.